

# CALIPSO Lidar Calibration at 1064 nm: Version 4 Algorithm

Mark Vaughan[1], Anne Garnier[1,2], Damien Josset[3], Melody Avery[1], Kam-Pui Lee[1,2], Zhaoyan Liu[1], William Hunt[1,2,†], Jacques Pelon[4], Yongxiang Hu[1], Sharon Burton[1], Johnathan Hair[1], Jason L. Tackett[1,2], Brian Getzewich[1,2], Jayanta Kar[1,2], and Sharon Rodier[1,2]

5 [1]NASA Langley Research Center, Mail Stop 475, Hampton VA USA 23681–2199
[2]Science Systems and Applications Inc. (SSAI), 1 Enterprise Parkway, Suite 200, Hampton, VA USA 23666
[3]US Naval Research Laboratory, NASA Stennis Space Center, MS USA 39529
[4]Laboratoire Atmosphères, Milieux, Observations Spatiales, UPMC-UVSQ-CNRS, Paris, France
[†]deceased

10 *Correspondence to*: Mark Vaughan (mark.a.vaughan@nasa.gov)

**Abstract.** Radiometric calibration of space-based elastic backscatter lidars is accomplished by comparing the measured backscatter signals to theoretically expected signals computed for some well-characterized calibration target. For any given system and wavelength, the choice of calibration target is dictated by several considerations, including signal-to-noise ratios (SNR) and target availability. This paper describes the newly implemented procedures used to calibrate the 1064 nm measurements acquired by CALIOP (i.e., the Cloud Aerosol Lidar with Orthogonal Polarization), the two-wavelength (532 nm and 1064 nm) elastic backscatter lidar currently flying on the Cloud Aerosol Lidar and Infrared Pathfinder Satellite Observations (CALIPSO) mission. CALIOP's 532 nm channel is accurately calibrated by normalizing the molecular backscatter from the uppermost aerosol-free altitudes of the CALIOP measurement region to molecular model data obtained from NASA's Global Modeling and Assimilation Office. However, because CALIOP's SNR for molecular backscatter measurements is prohibitively lower at 1064 nm than at 532 nm, the direct high-altitude molecular normalization method is not a viable option at 1064 nm. Instead, CALIOP's 1064 nm channel is calibrated relative to the 532 nm channel using the backscatter from a carefully selected subset of cirrus cloud measurements. In this paper we deliver a full account of the revised 1064 nm calibration algorithms implemented for the version 4.1 (V4) release of the CALIPSO lidar data products, with particular emphases on the physical basis for the selection of "calibration quality" cirrus clouds and on the new averaging scheme required to characterize intra-orbit calibration variability. The V4 procedures introduce latitudinally varying changes in the 1064 nm calibration coefficients of 25% or more relative to previous data releases and are shown to substantially improve the accuracy of the V4 1064 nm attenuated backscatter coefficients. By evaluating calibration coefficients derived using both water clouds and ocean surfaces as alternate calibration targets, and through comparisons to independent, collocated measurements made by airborne high spectral resolution lidar, we conclude that the CALIOP V4 1064 nm calibration coefficients are accurate to within 3 %.



## 1 Introduction

The Cloud Aerosol Lidar and Infrared Pathfinder Satellite Observations (CALIPSO) mission was launched on 28 April 2006 with a payload of three Earth-observing sensors: a single channel (645 nm) wide field-of-view camera, a three channel (8.65 μm, 10.6 μm, and 12.05 μm) imaging infrared radiometer (IIR), and the Cloud Aerosol Lidar with Orthogonal Polarization (CALIOP). CALIOP is an elastic backscatter lidar that transmits linearly polarized light at two wavelengths (532 nm and 1064 nm), and separately measures the total backscattered signal at 1064 nm and the components of the 532 nm backscattered signal polarized parallel and perpendicular to the polarization plane of the transmitted beam (Hunt et al., 2009). CALIPSO flies in a sun-synchronous orbit inclined at 98°, acquiring near-continuous measurements between 82° S and 82° N on a 16-day repeat cycle (Hunt et al., 2009). CALIOP acquired its first backscatter profiles on 7 June 2006 and has now delivered over 12 years of altitude-resolved measurements of clouds and aerosols in the Earth's atmosphere.

An essential precondition required to reliably derive the spatial and optical properties of clouds and aerosols from the CALIOP measurements (or from any other Earth-observing elastic backscatter lidar) is the accurate calibration of the measured backscatter data. In particular, accurate calibration of the CALIOP 1064 nm measurements is critically important in subsequent analyses such as reliably discriminating clouds from aerosols (Liu et al., 2018) and in retrieving accurate estimates of aerosol optical depths (Young et al., 2013; Young et al., 2016; Kim et al., 2018). To date, the radiometric calibration of space-based elastic backscatter lidar measurements has always been accomplished by calculating time-varying scale factors that provide the best near-instantaneous match between the measured data and the theoretically expected backscatter signals derived for some stable, well-characterized calibration target. The choice of calibration target depends critically on target availability and the signal-to-noise ratio (SNR) of the target measurements. By far the most common target is the Earth's atmosphere, which at very high altitudes is essentially free of aerosol contamination and hence the expected molecular backscatter can be well-characterized using the temperature and pressure profiles provided by atmospheric model data (e.g., from NASA's Global Modeling and Assimilation Office (GMAO)). The first space-based Earth-observing lidar, the Lidar In-space Technology Experiment (LITE), which flew aboard the space shuttle in September 1994 (Winker et al., 1996), used this high-altitude molecular normalization technique to calibrate the 355 nm and 532 nm measurements (Osborn, 1998; Osborn et al., 1998). However, because the molecular scattering cross-sections at 1064 nm are a factor of ~ 17 lower than at 532 nm (and ~ 89 times lower than at 355 nm), the 1064 nm SNR in the high-altitude calibration region precluded the use of the molecular normalization technique for those data. As a consequence, the 1064 nm measurements were left uncalibrated in LITE level 1 data distributed by NASA's Atmospheric Sciences Data Center (see https://eosweb.larc.nasa.gov/project/lite/lite_table).

Following the release of the LITE data, Reagan et al. (2002) devised a method to calibrate the 1064 nm channel using the backscatter signals from dense cirrus clouds. The ice crystal sizes within the clouds used by the calibration routines are assumed to be quite large with respect to the laser wavelengths, and hence the in-cloud extinction is concomitantly assumed





to be spectrally independent. Reagan et al. further argue that the cirrus backscatter coefficients are also spectrally independent at 532 nm and 1064 nm, and thus estimates of the 1064 nm calibration coefficients could be obtained by comparing the uncalibrated 1064 nm measurements to the calibrated 532 nm measurements of strongly scattering cirrus clouds. Reagan's technique was subsequently used to calibrate 1064 nm lidar measurements made by the Geoscience Laser

Altimeter System (GLAS), a two-channel space-based elastic backscatter instrument that launched on 12 January 2003 (Palm et al., 2004; Spinhirne et al., 2005).

The CALIOP 1064 nm calibration scheme also traces its lineage directly back to the pioneering work of Reagan et al. (2002). However, the use of cirrus clouds as a calibration target is not uniformly implemented for all space-based lidars. Unlike LITE, GLAS and CALIOP, the Cloud-Aerosol Transport System (CATS) used the molecular normalization

technique, together with estimates of aerosol loading provided by CALIOP, to calibrate their 1064 nm measurements over an altitude range of 23 to 27 km (Yorks et al., 2016). This was possible because the CATS instrument design delivered nighttime SNR in the low–to–middle stratosphere that was substantially higher than in the earlier systems. The CATS transmitters have laser pulse rate frequencies (PRFs) of 4 kHz and 5 kHz, per-pulse energies of 1–2 mJ, and are coupled to receivers that use photon counting detection at 1064 nm. In contrast, CALIOP has a PRF of 20.16 Hz, a nominal per-pulse

energy of 100–110 mJ, and detects the backscattered energy at 1064 nm using an avalanche photodiode (APD) (Hunt et al., 2009). While the APD has a relatively high quantum efficiency, it also has a high detector dark count rate, which contributes significant levels of noise in the high-altitude molecular signals. These high noise levels, combined with the greatly reduced sensitivity to molecular scattering, eliminate high altitude molecular normalization as a viable option for calibrating the CALIOP 1064 nm channel.

Through the course of three major data releases spanning ~ 8 years of on-orbit operations, the CALIOP 1064 nm calibration scheme remained relatively unchanged. The theoretical basis of the original algorithm is given in Hostetler et al. (2005). Vaughan et al. (2010) provide details on some procedural modifications that were incorporated for the version 3 (V3) data release. In contrast to these V3 updates, the version 4.1 (V4) release is a comprehensive upgrade that features major changes to all of the primary components of the 1064 nm calibration algorithm. In particular, we (a) defined a detailed set of sharply

focused criteria to identify a much more homogeneous population of clouds used in the calibration; (b) implemented a wholly new data averaging scheme that reduces uncertainties while simultaneously preserving intra-orbit variations in the calibration coefficients; and (c) augmented the lidar level 1 (L1) data products with substantially more robust estimates of calibration uncertainties. This paper describes all of these changes in detail. In doing so, we make repeated references to several earlier works. The fundamentals of the CALIOP 532 molecular normalization technique are given in Hostetler et al.

(2005) (hereafter H05) and Powell et al. (2009) (hereafter, P09). Initial development of CALIOP's 1064 nm cirrus cloud calibration scheme and the mathematical development of the error propagation is given in H05, with post-launch updates provided in Vaughan et al. (2010) (hereafter V10). The paper is organized as follows. Section 2 reviews the fundamental assumptions and equations that are used in the CALIOP 1064 nm calibration scheme. Section 3 provides a brief review of





the specific techniques used for the V3 data release and highlights the shortcomings that motivated the development of the V4 scheme. Details of the V4 approach, including the physical basis for selecting "calibration quality" cirrus clouds and the constraints involved in developing a multi-orbit data averaging scheme, are given in Sect. 4. An in-depth comparison between the V3 and V4 calibrations is conducted in Sect. 5, while Sect. 6 explores a variety of internal consistency checks

and validation techniques. Concluding remarks are given in Sect. 7.

## 2   CALIOP 1064 nm calibration fundamentals

The CALIOP 1064 nm calibration scheme uses the calibrated 532 nm attenuated backscatter coefficients measured in cirrus clouds to derive 1064 nm calibration coefficients from the simultaneously acquired but as yet uncalibrated 1064 nm cirrus measurements. As described in Sect. 3 of V10, in V3 and earlier the equation used to transfer the 532 nm calibration to the

1064 nm channel is

$$C_{1064} = f_{V3} \, C_{532} \,, \tag{1}$$

where $C_{1064}$ and $C_{532}$ are, respectively, the calibration coefficients at 1064 nm and 532 nm. $f_{V3}$ is a calibration scale factor computed using measurements acquired at both wavelengths:

$$f_{V3} = \chi_{cirrus}^{-1} \left( \frac{\langle X'_{1064}(z) \rangle}{\langle X'_{532}(z) \rangle} \right). \tag{2}$$

In this expression, $X'_{\lambda}(z)$ is the background-subtracted, range-corrected, gain and energy normalized measured backscatter signal at altitude z and wavelength λ (i.e., either 532 nm or 1064 nm), with additional corrections applied to account for molecular and ozone attenuations (V10). $\langle X'_{\lambda}(z) \rangle$ is the mean value of $X'_{\lambda}(z)$, computed from cloud top to cloud base. In terms of the atmospheric components being measured,

$$X'_{\lambda}(z) = C_{\lambda} \left( \beta_{\lambda,p}(z) + \beta_{\lambda,m}(z) \right) T^2_{\lambda,p}(z) \,, \tag{3}$$

where $C_{\lambda}$ is the wavelength-dependent calibration coefficient, $\beta_{\lambda}(z)$ is the wavelength-dependent backscatter coefficient for either particulates (subscript p) or molecules (subscript m), and $T^2_{\lambda,p}(z)$ is the particulate two-way transmittance due to the ice crystals in cirrus clouds. Because these crystals are quite large relative to the CALIOP wavelengths, the extinction coefficients are assumed to be spectrally independent, and hence





$$\frac{X'_{1064}(z)}{X'_{532}(z)} = \left(\frac{C_{1064}}{C_{532}}\right)\left(\frac{\beta_{1064,p}(z)+\beta_{1064,m}(z)}{\beta_{532,p}(z)+\beta_{532,m}(z)}\right).$$

(4)

The remaining term in Eq. (2), $\chi_{cirrus}$, is the mean backscatter color ratio for cirrus clouds, defined as

$$\chi_{cirrus} = \frac{\langle \beta_{p,1064}(z)\rangle}{\langle \beta_{p,532}(z)\rangle},$$

(5)

where the angle brackets once again represent the mean value computed from cloud top to cloud base. Note that when Eq.

(4) is applied to appropriately selected cirrus clouds, $\beta_{1064,p}(z) \approx \beta_{532,p}(z)$ (i.e., the assumption invoked by Reagan et al. (2002)) and $\beta_{532,p}(z) \gg \beta_{532,m}(z)$, so that the ratio of the total backscatter coefficients (i.e., molecular and particulate combined) reduces to a very close approximation to the ratio of the particulate backscatter terms alone.

On the right-hand side of Eq. (1), the values of $\langle X'_{532}(z)\rangle$ and $\langle X'_{1064}(z)\rangle$ used to compute $f_{V3}$ are obtained directly from the measured data, and $C_{532}$ is derived by calibrating the 532 nm data (Kar et al., 2018; Getzewich et al., 2018). The $\chi_{cirrus}$ term

in $f_{V3}$ is an externally prescribed a priori value, and the only quantity in Eq. (1) that is not directly derived from CALIOP's onboard measurements. For versions 1 and 2 of the CALIPSO data products, $\chi_{cirrus}$ was assumed to be $1.00 \pm 0.04$ (H05). This assumption was revisited and revised prior to the release of V3. Based on the analysis of over 400 hours of multi-wavelength elastic backscatter measurements acquired by the Cloud Physics Lidar (McGill et al., 2002), $\chi_{cirrus}$ is now assigned a uniform value of $1.01 \pm 0.25$ for all CALIOP 1064 nm calibration procedures. Our rationale for this choice is

described at length in V10. Recent field observations using the Raman lidar technique at both 532 nm and 1064 nm provide further evidence for the spectral independence of cirrus backscatter (Haarig et al., 2016), as do previous elastic backscatter lidar measurements acquired at 550 nm and 728 nm (Ansmann et al., 1993) and multi-wavelength Raman measurements acquired at 355 nm and 532 nm (Beyerle et al., 2001).

The uncertainties in the V3 CALIOP 1064 nm calibration coefficients are estimated using

$$\left(\frac{\Delta C_{1064}}{C_{1064}}\right)^2 = \left(\frac{\Delta\chi_{cirrus}}{\chi_{cirrus}}\right)^2 + \left(\frac{\Delta\langle X'_{1064}(z)\rangle}{\langle X'_{1064}(z)\rangle}\right)^2 + \left(\frac{\Delta\langle X'_{532}(z)\rangle}{\langle X'_{532}(z)\rangle}\right)^2 + \left(\frac{\Delta C_{532}}{C_{532}}\right)^2,$$

(6)

where $\Delta a$ is the standard deviation or random uncertainty in the quantity $a$ (see Eq. (12) in V10). Because the 1064 nm calibration coefficients are not independent calculations, but are instead derived from previous calibrations of the 532 nm channel, the uncertainties in the 1064 nm calibration coefficients depend directly on the uncertainties estimated for the 532 nm calibration coefficients. The relationships between the different components of the calibration procedure are diagramed



in Figure 1. The 1064 nm daytime calibration coefficients are derived from the 532 nm daytime calibration coefficients, which in turn are derived from the 532 nm nighttime calibration coefficients (Getzewich et al., 2018). The uncertainties for the daytime 1064 nm daytime calibration coefficients thus contain contributions from both the daytime and nighttime 532 nm calibrations.

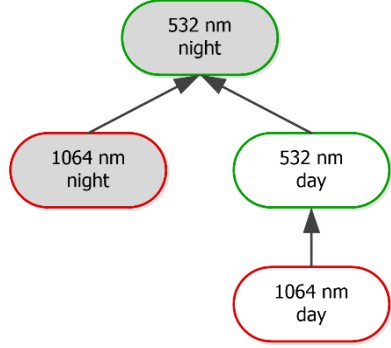

**Figure 1: Dependency diagram showing the functional relationships between the 532 nm and 1064 nm CALIOP calibration algorithms**

## 3    Motivation for change: the V3 calibration algorithms

Effective use of the 1064 nm calibration equation implicitly requires two critically important subroutines: (1) an algorithm to
identify cirrus clouds appropriate for use in the calibration calculation, and (2) a data averaging scheme to reduce the random noise in the calibration coefficients without simultaneously introducing biases. In this section we review the methods used for these tasks in the CALIOP V3 calibration scheme and point out the shortcomings that led to the subsequent development of revised techniques for V4.

### 3.1    Identifying calibration quality cirrus

As described in H05 and V10, the technique for identifying the calibration targets (i.e., clouds) used in the V3 calibration scheme is straightforward. First, profiles of 532 nm attenuated backscatter coefficients, $\beta'_{532}(z)$, are averaged horizontally and then converted to profiles of attenuated scattering ratios, $R'_{532}(z)$, where

$$R'_\lambda(z) = \frac{\left(\beta_{\lambda,m}(z)+\beta_{\lambda,p}(z)\right)T^2_{\lambda,m}(z)T^2_{\lambda,O_3}(z)T^{2\eta}_{\lambda,p}(z)}{\beta_{\lambda,m}(z)T^2_{\lambda,m}(z)T^2_{\lambda,O_3}(z)} = \left(1+\frac{\beta_{\lambda,p}(z)}{\beta_{\lambda,m}(z)}\right)T^{2\eta}_{\lambda,p}(z).\tag{7}$$

The numerator of Eq. (7) represents the measured attenuated backscatter coefficients, where $\beta_{\lambda,x}(z)$ is a backscatter
coefficient measured for constituent x at altitude z and wavelength λ. The constituent-specific attenuations are given by the two-way transmittance terms,



$$T_{\lambda,x}^{2\eta}(z) = \exp\left( -2\eta_{\lambda,x} \int_{z_0}^{z} \sigma_{\lambda,x}(r)\,dr \right), \tag{8}$$

where $\sigma_{\lambda,x}(z)$ is an extinction coefficient and $\eta_{\lambda,x}$ is a layer-effective multiple scattering factor. The subscripts $O_3$ and m indicate contributions from, respectively, ozone absorption and molecular backscatter and attenuation, and $\eta_m = \eta_{O_3} = 1$ (Winker, 2003; Young and Vaughan, 2009). The quantities in the denominator of Eq. (7) are derived from meteorological

model data (i.e., profiles of molecular and ozone number densities) obtained from the GMAO. Nighttime profiles of $R'_{532}(z)$ are averaged over 15 consecutive laser pulses (~ 5 km along-track). To reduce the additional noise introduced by solar background signals, daytime profiles are averaged over 30 consecutive laser pulses (~ 10 km along-track). These attenuated scattering ratio profiles are then searched downward over an altitude range from 17 km to 8.2 km in order to identify the highest altitude for which $R'_{532}(z) > 50$ for three or more consecutive range bins. All regions satisfying this

search criterion are identified as calibration quality clouds and subsequently used in the V3 1064 nm calibration calculations. Requiring the scattering ratio to exceed 50 throughout the layer minimizes calibration biases by ensuring that the molecular contributions to the total backscatter signals will be negligible (see Sect. 7 in H05). On the other hand, the V3 L1 requirements for identifying a layer are very different from those used in the V3 level 2 (L2) analysis, and hence V3 calibration quality clouds typically appear as strongly scattering regions embedded within the more vertically extensive

structures reported in the V3 L2 data products.

While the V3 L1 detection scheme effectively identifies strongly scattering cloud regions between 17 km and 8.2 km, it is also subject to three kinds of sampling bias: suboptimal sampling as a function of latitude, due to fixed altitude limits; contamination by water clouds in the tropics, and/or by polar stratospheric clouds (PSCs) in the polar regions; and differential attenuation of the backscatter signals due to undetected layers lying above the top of the calibration cloud. The

first two of these effects are illustrated in Figure 2, which shows the zonal mean occurrence frequency for ice clouds detected at night in the V4 level 2 data during August 2016. Between ~ 20° S and ~ 30° N, the predefined V3 search limits encompass ~ 90 % of all range bins classified as containing ice. However, as seen in Figure 3, outside of this latitude range, the fraction of ice clouds falling with the V3 search limits drops linearly, falling to less than 50 % at ~ 34° S and ~ 54° N. Between ~ 70° S and ~ 50° S, approximately 75 % of the potential 'calibration quality' clouds – i.e., tropospheric cirrus – are

located below the minimum search altitude of 8.2 km. Based solely on Figure 3, the fraction of clouds available as potential calibration targets appears to increase to ~ 50 % poleward of ~ 70° S. However, Figure 2 shows that this apparent increase is illusory, as this region is dominated by PSCs that lie well above the local tropopause altitude. The particle sizes in PSCs are often substantially smaller than is typical for tropospheric cirrus (Reichardt et al., 2004; Heymsfield et al., 2014), and thus the requisite assumption that $\chi_{cirrus} \approx 1$ cannot be confidently applied for these layers.





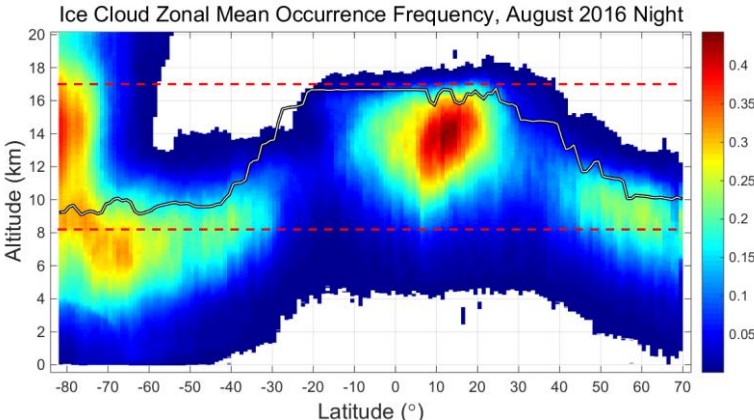

**Figure 2: zonal mean occurrence frequency of ice clouds for V4 nighttime data acquired during August 2016. The solid gray line shows the mean tropopause heights for the month, while the red dashed lines demarcate the V3 calibration cloud search region between 17 km and 8.2 km. Polar stratospheric clouds are responsible for the high occurrence frequencies above the tropopause poleward of ~ 60° S.**

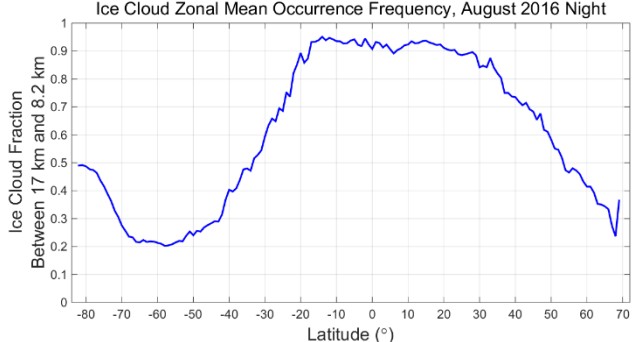

**Figure 3: latitudinally varying fraction of ice cloud range bins in Figure 2 that lie within the V3 calibration cloud search limits.**

The V3 calibration cloud identification scheme relies solely on the magnitude of the attenuated scattering ratios within a fixed altitude range, and does not consider other available information such as volume depolarization ratios and/or in-cloud temperatures. One consequence of this choice is the introduction of the second of the three kinds of sampling bias: ~ 13% of the clouds used in the V3 calibration scheme are almost certainly water, not ice. This is illustrated by Figure 4a, which shows the occurrence frequency of the layer-integrated volume depolarization ratios, $\delta_v$, for all V3 calibration clouds identified during August 2013. The distribution is clearly bimodal, with a primary peak at $\delta_v \approx 0.39$, consistent with cirrus cloud depolarization (Sassen et al., 2012), and a secondary peak at $\delta_v \approx 0.10$, consistent with the multiple scattering induced depolarization observed by CALIOP in dense water clouds (Hu, 2007). Figure 4b shows the distribution of $\delta_v$ as a function of mean $R'_{532}$ for the same V3 calibration clouds. Depolarization ratios below 0.2 are seen to increase approximately linearly as a function of $R'_{532}$, as is expected for increasingly dense liquid water clouds (Hu, 2007). On the other hand, there is no obvious trend for those clouds having $\delta_v > \sim 0.3$. Figure 4c shows the distribution of $\delta_v$ as a function of mid-layer





temperature ($T_{mid}$). The depolarization ratios less than 0.2 are strongly associated with warmer temperatures, giving further credence to the supposition that these clouds are supercooled water clouds.

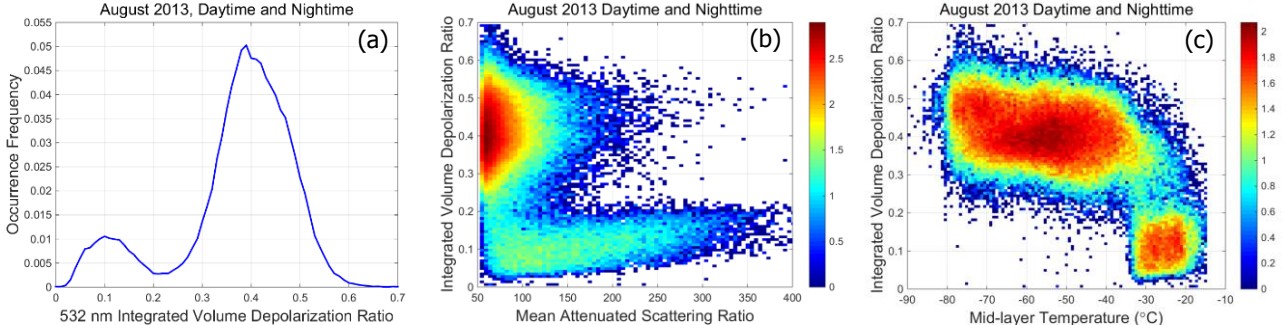

**Figure 4: panel a (left) shows the distribution of layer-integrated volume depolarization ratios for all calibration clouds identified by the V3 algorithm during August 2013; panel b (center) shows the distribution of layer-integrated volume depolarization ratios as a function of layer mean attenuated scattering ratio; and panel c (right) shows the distribution of layer-integrated volume depolarization ratios as a function of mid-layer temperature. The colors in panels a and c indicate log₁₀ of the number of samples per bin.**

The third type of bias occasioned by the V3 calibration routine is the risk of differential attenuation of the 532 nm and 1064 nm signals. While $X'_{532}$ and $X'_{1064}$ are both corrected for wavelength-dependent attenuation effects due to molecular and ozone two-way transmittances, at this initial stage of the lidar data analysis, no correction is possible for as-yet-undetected particulates (i.e., cloud or aerosol layers) lying between the lidar and the top of the calibration cloud. A more rigorous expansion of Eq. (2) would explicitly include these terms; i.e.,

$$f_{V3} = \chi_{cirrus}^{-1} \left( \frac{T_{p,1064}^2\left(0, r\left(z_{top}\right)\right) \left\langle X'_{1064}\left(r\left(z_{top}\right), r\left(z_{base}\right)\right)\right\rangle}{T_{p,532}^2\left(0, r\left(z_{top}\right)\right) \left\langle X'_{532}\left(r\left(z_{top}\right), r\left(z_{base}\right)\right)\right\rangle} \right), \tag{9}$$

where $T_{p,\lambda}^2\left(0, r\left(z_{top}\right)\right)$ represents the particulate two-way transmittance between the lidar (at range = 0) and the top of the calibration cloud (at range = $z_{top}$) and the mean signals are explicitly calculated over the range from $z_{top}$ to $z_{base}$. The ubiquitous presence of stratospheric aerosols suggests that, because the stratospheric extinction and aerosol optical depth (AOD) are typically larger at 532 nm than at 1064 nm (Thomason and Peter, 2006), $f_{V3}$ is slightly overestimated because, in general, $T_{p,1064}^2\left(0, r\left(z_{top}\right)\right) \big/ T_{p,532}^2\left(0, r\left(z_{top}\right)\right) > 1$. For the most part, this kind of bias error is negligible. However, on those occasions when substantial aerosol or PSC layers are located above a V3 calibration cloud, the resulting biases in $f_{V3}$ can be significant (e.g., $T_{p,1064}^2/T_{p,532}^2 \approx 1.25$ and higher at the tops of clouds located below the Black Saturday smoke plumes over Australia in February 2009).





## 3.2    The V3 calibration averaging scheme

Although individual estimates of $f_{V3}$ use high SNR measurements (i.e., $R'_{532}(z) > 50$), the uncertainties for these estimates are still large, and thus obtaining reliable values requires some amount of signal averaging. To maximize the number of $f_{V3}$ samples averaged, the V3 scheme computes mean values of $f_{V3}$, denoted as $\langle f_{V3} \rangle$, over each *granule* of the CALIOP data

record (H05, V10). CALIOP data granules extend from one terminator to the next, thus dividing each orbit into separate daytime and nighttime segments. This averaging scheme implicitly assumes that the pattern of thermally driven intra-orbit changes observed in the 532 nm calibration coefficients (P09) is reproduced more-or-less identically in the 1064 nm calibration coefficients, and hence $f_{V3}$ can be considered constant with respect to the elapsed time throughout the individual daytime and nighttime segments of any orbit. The time-varying V3 1064 nm calibration coefficients are then computed

using $C_{1064}(t) = \langle f_{V3} \rangle C_{532}(t)$, where t represents granule-elapsed time and $C_{532}(t)$ is the 532 nm calibration coefficient at time t. As illustrated in Figure 5, monthly averages of instantaneous calculations $f_{V3}$, computed as functions of granule-elapsed time and plotted as functions of latitude, demonstrate conclusively that the assumption that $f_{V3}$ is constant within a granule is not valid. $f_{V3}$ is seen to exhibit a strong dependence on granule elapsed time, and can vary by up to 40 % or more within a single granule. Furthermore, $f_{V3}$ exhibits a seasonally varying hysteresis, with latitudinal day–night differences

being maximized in the boreal winter (Figure 5a) and minimized during the boreal summer (Figure 5c).



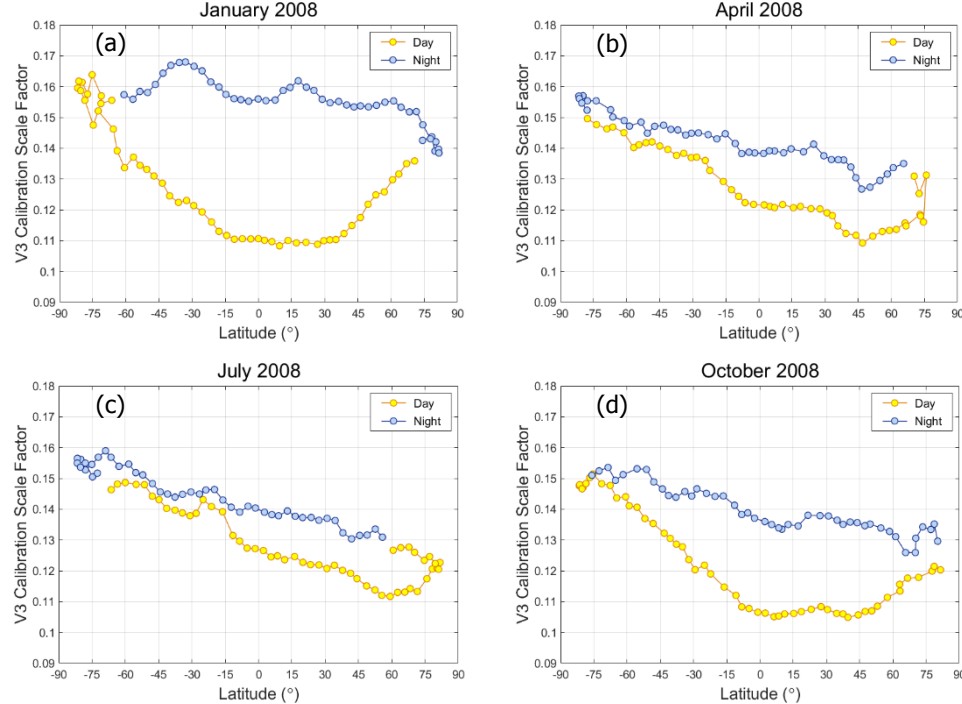

**Figure 5: monthly averages of daytime (yellow) and nighttime (blue) V3 calibration scale factors (i.e., $f_{V3}$) as computed as functions of granule-elapsed time and plotted as functions of latitude for (a) January, (b) April, (c) July, and (d) October 2008.**

While the underlying causes of the time-varying behaviors of $f_{V3}$ have not yet been determined, accurately compensating for these changes remains essential for reliably calibrating the CALIOP 1064 nm measurements. Revising the averaging scheme to compute running averages of $f_{V3}$ as a function of granule-elapsed time would seem to be an obvious strategy for characterizing the intra-orbit changes observed in Figure 5. However, successful application of this approach on a single granule basis is unlikely, simply because the occurrence of a sufficient number of calibration quality clouds at any location
or within any time frame cannot be guaranteed.

Figure 6 (from Vaughan et al., 2012) shows the monthly occurrence frequency of V3 calibration quality clouds detected during daytime granules as a function of granule elapsed time (y-axis) for each calendar month from June 2006 through December 2010 (x-axis). The white grid cells seen along the top edge of the figure represent regions where no suitable clouds were detected for the entire month. The sample counts throughout the tropics (i.e., the oscillating dark red region
between elapsed times of ~ 1100 s to ~ 2100 s) are always quite high, and hence estimates of $f_{V3}$ can be readily obtained in this region. However, sample counts in the Arctic (elapsed time > 2500 s) during spring 2008 or late winter 2009 are extremely low, and the likelihood of obtaining trustworthy estimates of $f_{V3}$ in these times and places is likewise extremely low. Clearly then, any new averaging scheme devised for the V4 calibration must simultaneously accomplish two tasks. First, it must characterize the calibration scale factors as a function of granule elapsed time throughout the full extent of each





granule. And second, in order to produce high SNR estimates of these time-varying scale factors, the new averaging scheme, in concert with the revised cloud selection routine, must harvest significantly more calibration quality clouds at all latitudes than would be available using the V3 algorithm.

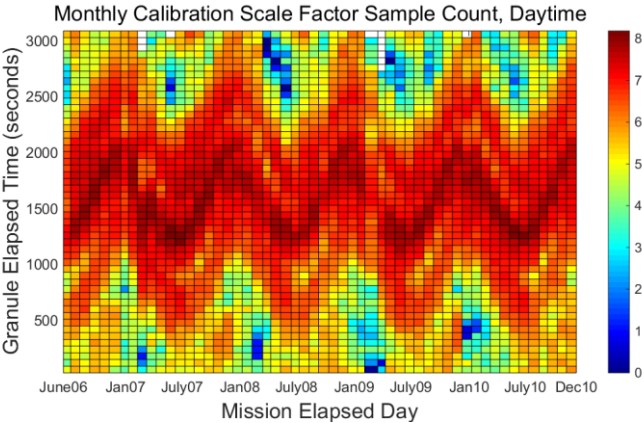

**Figure 6: monthly counts of V3 daytime scale factor calculations as a function mission elapsed time (x-axis) and granule elapsed time (y-axis). Colors are displayed on a $\log_{10}$ scale, so that dark reds indicate many thousands of samples, whereas dark blues indicate one or two samples. Regions where no calibration quality clouds were detected are shown in white.**

### 3.3 Calculating profiles of V3 attenuated backscatter coefficients

Once $\langle f_{V3} \rangle$ has been computed for a granule, the V3 1064 nm calibration coefficients are computed for each profile in the granule using Eq. (1). The altitude-resolved profiles of 1064 nm attenuated backscatter, $\beta'_{1064}(z)$, reported in the CALIPSO lidar level 1 data products are then derived as follows (H05):

$$\beta'_{1064}(z) = \frac{r(z)^2 \left( P_{1064}(z) - P_{1064,bkg} \right)}{G_{1064}\, E_{1064}\, C_{1064}} = \left( \beta_{1064,m}(z) + \beta_{1064,p}(z) \right) T^2_{\lambda,m}(z)\, T^2_{\lambda,O_3}(z)\, T^{2\eta}_{\lambda,p}(z), \tag{10}$$

where $P_{1064}(z)$ is backscattered signal from altitude z measured aboard the satellite in the 1064 nm receiver (units = digitizer counts), $P_{1064,bkg}$ is the background signal measured aboard the satellite for each profile, and r(z) is the range (units = km) from the lidar to altitude z. $E_{1064}$ is the per-pulse energy transmitted at 1064 nm (units = J) and $G_{1064}$ quantifies the electronic gain at 1064 nm (unitless). The subscripts m, p, and $O_3$ once again indicate contributions from, respectively, molecules, particulates, and ozone. The units of $\beta'_{1064}(z)$ are $km^{-1}\, sr^{-1}$. The units of $C_{1064}$ are $km^3 \times sr \times J^{-1} \times counts$.

### 4 The version 4 calibration algorithms

To correct the shortcomings discovered in the V3 calibration scheme, CALIOP's V4 algorithm differs from its predecessors in three fundamental aspects: cirrus cloud selection, data averaging, and the characterization of uncertainties. Each of these




will be addressed in the following subsections. Additionally, we incorporate a seemingly small, but nonetheless important, change in the way the V4 calibration scale factors, $f_{V4}$, are calculated. The same calibration transfer equation still applies; i.e., $C_{1064} = f_{V4} C_{532}$, as in Eq. (1). However, in computing $f_{V4}$, the layer-mean values of the background-subtracted, range-corrected, gain and energy normalized measured backscatter signals, $\langle X'_\lambda(z)\rangle$, are replaced with the integrated values, $g_\lambda$,

where

$$g_\lambda = \int_{top}^{base} X'_\lambda(r)\,dr - dg_\lambda, \text{ where } dg_\lambda = \frac{1}{2}\left(z_{top} - z_{base}\right)\left(X'_\lambda\left(z_{base}\right) + X'_\lambda\left(z_{top}\right)\right) \tag{11}$$

(e.g., as derived in equations 18–20 in V10), and thus

$$f_{V4} = \chi_{cirrus}^{-1}\left(g_{1064}\Big/g_{532}\right). \tag{12}$$

The $dg_\lambda$ terms represent corrections for the molecular scattering contributions to the signals measured within the cloud
boundaries. As explained in detail in Sect. 4.1, the V4 cirrus cloud selection method no longer enforces the large scattering ratio requirement ($R'_{532} > 50$) that allowed us to neglect these contributions in V3, and thus corrections for molecular scattering are essential in the V4 calibration algorithm. Note, though, that the correction is only applied at 532 nm. Because CALIOP is largely insensitive to molecular scattering at 1064 nm, $dg_{1064}$ is set uniformly to zero.

### 4.1 Selecting calibration quality cirrus clouds

The selection of calibration quality clouds in V3 was based on two globally applied criteria: layer altitude and the magnitude of $R'_{532}(z)$. In contrast, the V4 algorithm identifies calibration quality clouds based on four different quantities: layer altitude, mid-layer temperature ($T_{mid}$), layer integrated volume depolarization ($\delta_v$), and layer integrated attenuated backscatter at 532 nm ($\gamma'_{532}$). These latter two quantities are defined as, respectively,

$$\delta_v = \frac{\sum_{j=top}^{base} X_\perp(z_j)}{\sum_{j=top}^{base} X_\parallel(z_j)}, \tag{13}$$

where $X_\perp(z)$ and $X_\parallel(z)$ are, respectively, the signals measured at altitude z in the 532 nm perpendicular and parallel channels, and



$$\gamma'_\lambda = \frac{g_\lambda}{C_\lambda} = \int_{z_{\text{top}}}^{z_{\text{base}}} \beta'_\lambda(r)\,\mathrm{d}r - \mathrm{d}\beta'_\lambda, \tag{14}$$

where $\beta'_\lambda(z)$ is the attenuated backscatter coefficient at altitude z and wavelength λ and $\mathrm{d}\beta'_\lambda = \mathrm{d}g_\lambda / C_\lambda$.

### 4.1.1    V4 layer detection and selection based on altitude

The V3 calibration algorithm implemented a dedicated layer detection scheme that was sensitive only to strongly scattering
features. Moreover, as discussed in section 3.1, the fixed altitude range over which this layer detection procedure was applied effectively eliminated a large fraction of potential calibration quality clouds, while at the same time permitting the inclusion of PSCs for which the assumption of $\chi_{\text{cirrus}} \approx 1$ is not well-founded (Sect. 3.1). V4 addresses these defects in two ways. In the most far-reaching change, V4 abandons the dedicated layer detection scheme used in V3 and replaces it with the same layer detection algorithm that is used in the CALIOP L2 analyses (Vaughan et al., 2009). The L2 layer detection
algorithm identifies layers having a much wider range of backscatter intensity, and its cirrus detection capabilities have been extensively validated (McGill et al., 2007; Thorsen et al., 2011; Yorks et al., 2011; Candlish et al., 2013; Kim et al., 2014). In its standard configuration, the L2 layer detection algorithm applies a nested, multi-resolution data averaging scheme that detects layers at five different horizontal averaging resolutions: 1/3 km (i.e., single shot resolution), 1 km, 5 km, 20 km and 80 km. In the 1064 nm calibration algorithm, only the 5 km resolution is used, and thus, unlike V3, the profiles of $R'_{532}(z)$
used in the V4 layer detection algorithm are averaged uniformly over 15 consecutive shots for both daytime and nighttime analyses. These 5 km averaged profiles are then scanned between 30 km and the local surface altitude obtained from a digital elevation model (DEM) (Tanelli et al., 2014). Only the uppermost layer detected is further evaluated as a potential calibration quality cloud; layers detected at lower altitudes are discarded, irrespective of their scattering intensity. Enforcing this condition contributes to reducing the severity of the bias errors that can creep into the calculation of $f_{V4}$.

The second altitude-based change to the layer acceptance criteria is that the cirrus selection region is no longer static. Instead, within each 5 km horizontal average, a valid cirrus acceptance region is dynamically defined based on maximum altitudes of the local tropopause (obtained from GMAO atmospheric model data) and the Earth's surface (obtained from a DEM). To account for overshooting cloud tops and uncertainties in the tropopause height, the search for calibration quality clouds begins 2 km above the maximum GMAO tropopause altitude. Similarly, to eliminate the possibility of surface
contamination the search is terminated 1 km above the maximum DEM altitude.

These two changes have important consequences for the eventual selection of calibration quality clouds. This is illustrated in Figure 7a, which shows a smoke plume from the February 2009 Black Saturday fires in Australia (de Laat et al., 2012) that partially overlies an opaque cirrus cloud layer. As seen in Figure 7b, the attenuated scattering ratios in the cirrus below the smoke exceed the calibration quality cloud threshold implemented in the V3 algorithm, and thus this cloud was used to



calculate estimates of $f_{V3}$ in the V3 data set. But because smoke is strongly absorbing at 532 nm, with Ångström exponents typically in the neighborhood of 1.8–2.0 (Chand et al., 2006; Chand et al., 2008), the differential attenuation term in Eq. (9) becomes notably larger than 1, and the estimates of $f_{V3}$ are biased correspondingly high. This is not an issue in V4. The cirrus layer will not be considered for the calibration routine, simply because it is not the highest layer detected in the profile.

And while the smoke layer is considered, it is subsequently rejected based on additional criteria described in the following sections.

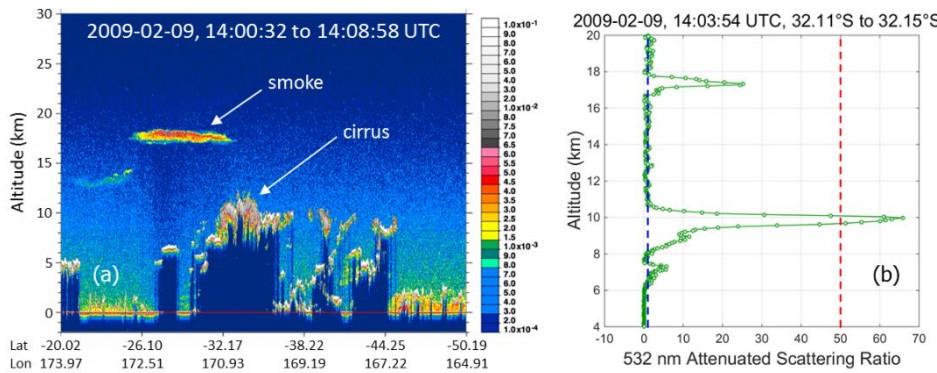

**Figure 7: (a) CALIOP 532 nm attenuated backscatter coefficients (km⁻¹ sr⁻¹) showing smoke from the February 2009 Black Saturday fires in Australia lofted over an opaque cirrus deck; (b) a profile of attenuated scattering ratios (in green) for which the**
**cirrus beneath the smoke plume qualifies as a calibration quality cloud in the V3 algorithm. In panel b, the blue dashed vertical line indicates an attenuated scattering ratio of 1, while the red dashed vertical line indicates the V3 cloud detection threshold of $R'_{532} = 50$. Below the high-altitude smoke plume, the ratio of particulate two-way transmittances is $T^2_{p,1064}/T^2_{p,532} = 1.25 \pm 0.20$.**

### 4.1.2   Selection based on $T_{mid}$ and $\delta_v$

In deriving a more comprehensive set of selection criteria for identifying calibration quality clouds, our initial efforts focused
on determining appropriate thresholds for mid-layer temperature and layer-integrated depolarization ratio. To test the proposition that supercooled water clouds (i.e., as in Figure 4) were biasing calculations of $f_{V3}$, we generated two months of test data (February and March, 2009) for which the layer search region was defined by the local tropopause and DEM surface (see Sect. 4.1.1), but the sole layer selection criterion remained, as in V3, $R'_{532} > 50$ for three consecutive range bins. As expected, changing the search region greatly increased the number of calibration quality clouds detected at higher
latitudes (red and black lines in Figure 8a). At the same time, this change also greatly increased both the mean magnitude of the calibration scale factors computed poleward of ±30° (red and black lines in Figure 8b) and the variability of the calibration scale factors computed in these regions. This increase in magnitude and variability is caused by the much wider range of mid-cloud temperatures in the lower altitude data set. When the test data are restricted to calibration clouds with mid-layer temperatures of −35° C or colder (blue and green lines in Figure 8), the number of samples poleward of ±30° falls
by an order of magnitude or more, and the scale factors drop to levels similar to those in the V3 data. Figure 9a shows the distribution of the scale factors as a function of mid-layer temperature. The scale factors appear to be naturally partitioned





into two clusters that fall on either side of a dividing line at –35° C, with the colder clouds having a lower mean scale factor and showing less variability.

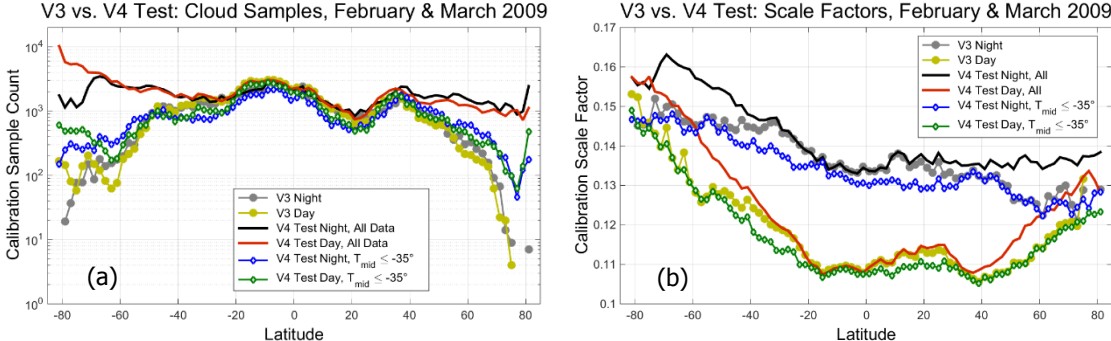

**Figure 8:** (a) sample counts and (b) mean scale factors for all daytime and nighttime granules acquired during February and March, 2009. V3 results are shown in yellow (day) and dark gray (night). The initial test results (new altitude regime only) are shown in red (day) and black (night). The test results with a –35° C temperature requirement imposed are shown in green (day) and blue (night).

As seen in Figure 9b, the 532 nm layer-integrated volume depolarization ratios also appear to cluster into two distinct groups, with centers falling on either side of a dividing line at $\delta_v = 0.3$. Figure 9c plots the occurrence frequency of $\delta_v$ as a function of $T_{mid}$, and shows a structure that is essentially identical to what is seen in Figure 4c. The dividing lines at $T_{mid} = -35°C$ and $\delta_v = 0.3$ partition the data into four quadrants. The upper left quadrant, where $T_{mid} < -35°C$ and $\delta_v > 0.3$ can be confidently assumed to contain only ice clouds (Campbell et al., 2015). The bottom right quadrant is, in all likelihood, populated mostly by supercooled water clouds. Table 1 shows the descriptive statistics for the scale factors associated with the data points in the upper left and lower right quadrants of Figure 9c. In the mean, the scale factors in the upper left quadrant are smaller than those in the lower right quadrant by ~ 19 %.

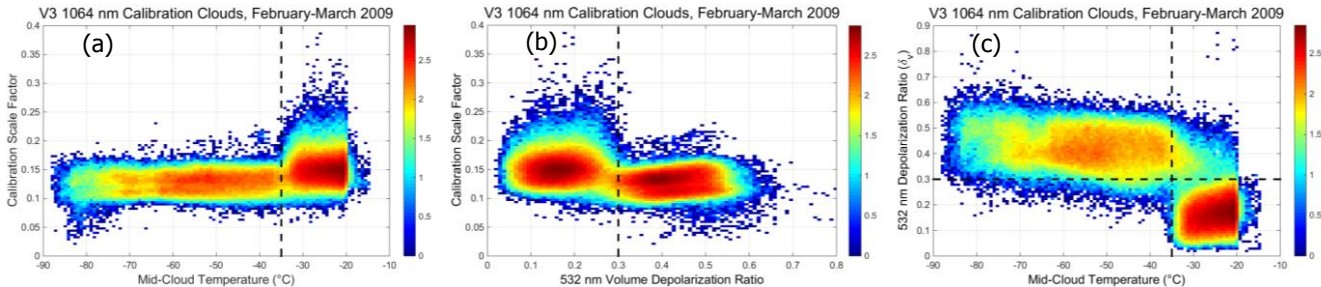

**Figure 9:** for the February and March 2009 test data set, (a) shows the occurrence frequency of $f_{V3}$ as a function of mid-layer temperature; (b) shows the occurrence frequency of $f_{V3}$ as a function of 532 nm layer-integrated volume depolarization ratio; and (c) shows the occurrence frequency of layer-integrated depolarization as a function of mid-layer temperature. For all panels, the plot colors represent $\log_{10}$ of the number of sample counts in each grid cell.





**Table 1: descriptive statistics for the scale factors associated with the data points in the upper left and lower right quadrants of the right panel in Figure 9 (MAD = median absolute deviation)**

|  | $T_{mid} < -35°C$ and $\delta_v > 0.3$ | $T_{mid} > -35°C$ and $\delta_v < 0.3$ |
|---|---|---|
| Minimum | 0.0174 | 0.0783 |
| Maximum | 0.2268 | 0.3979 |
| Median | 0.1270 | 0.1482 |
| MAD | 0.0129 | 0.0159 |
| Mean | 0.1262 | 0.1498 |
| Standard deviation | 0.0160 | 0.0214 |
| Samples | 104728 | 117176 |

### 4.1.3    Selection based on $\gamma'_{532}$

The fundamental assumption underlying the CALIOP 1064 nm calibration scheme is that because the ice crystals in cirrus
clouds are most often quite large relative to the CALIOP wavelengths, the layer-mean cirrus backscatter coefficients are
spectrally independent at 532 nm and 1064 nm (Reagan et al., 2002). Satisfying this assumption thus requires some method
for estimating cirrus particle size prior to calibrating the 1064 nm channel. To accomplish this, we used the CALIPSO V3
level 2 lidar and IIR track data products to derive an empirical relationship between $\gamma'_{532}$, which is readily obtained from the
calibrated 532 nm measurements, and the effective diameters retrieved from perfectly collocated IIR measurements (Garnier
et al., 2012; Garnier et al., 2013). Figure 10 compares the lidar measurements to the collocated IIR retrievals for all clouds
used in the V3 1064 nm calibration scheme during October 2010. As seen in Figure 10a, the V3 attenuated backscatter color
ratios, $\chi'_{layer} = \gamma'_{1064}/\gamma'_{532}$ , remain relatively constant for IIR effective diameters above ~ 35 μm, with a mean value of 0.96 ±
0.05. Similarly, Figure 10b shows that the majority of these large effective diameters are concentrated within a $\gamma'_{532}$ range
between 0.023 sr$^{-1}$ and 0.038 sr$^{-1}$. (Note that, consistent with the analyses in V10, both $\chi'_{layer}$ and $\gamma'_{532}$ are computed for the
entire cloud, and not just the strongly scattering region used in the V3 calibration procedure.) From this analysis, we
conclude that, for the V4 calibration scheme, we can isolate the population of large-particle cirrus for which $\chi'$ is relatively
constant by imposing the appropriate limits on $\gamma'_{532}$. Assuming a lidar ratio of 30 sr and a multiple scattering factor of 0.6
(Young et al., 2018), these limits on $\gamma'_{532}$ ensure that the optical depths of the clouds used in the calibration procedure are
typically larger than ~1.47.



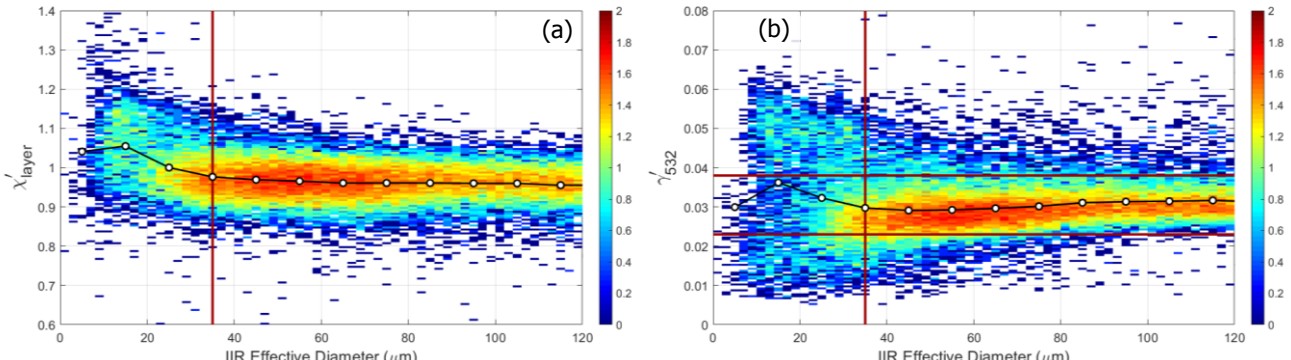

**Figure 10: (a)** $\chi'_{layer}$ **and (b)** $\gamma'_{532}$ **as functions of IIR-derived effective particle size for all nighttime calibration quality clouds detected by the V3 1064 nm calibration scheme during October 2010. The filled circles in each panel represent median values of the distributions.**

In addition to identifying clouds comprised of large particles, the V4 calibration cloud selection scheme must also ensure that these large particles are ice. For CALIOP, cloud ice-water phase is readily determined by the relationship between $\gamma'_{532}$ and $\delta_v$ (Hu, 2007; Hu et al., 2009). Figure 11 shows the joint occurrence frequencies of $\delta_v$ and $\gamma'_{532}$ for different subsets of clouds detected during October 2010. Figure 11a shows data from only those clouds that were detected at a 5-km horizontal resolution and were the highest cloud detected in each profile. Randomly oriented ice (ROI) clouds are characterized by smaller integrated attenuated backscatters and higher depolarization ratios, with $\delta_v$ for ice clouds being largely independent of $\gamma'_{532}$. Water clouds, on the other hand, generally have much larger integrated attenuated backscatter coefficients, and there is a strong linear relationship between the magnitudes of $\delta_v$ and $\gamma'_{532}$. The small population of clouds dominated by horizontally oriented ice (HOI) crystals, shown in the bottom right of Figure 11a, has very large $\gamma'_{532}$ and $\delta_v$ close to zero. Figure 11b shows $\delta_v$ and $\gamma'_{532}$ calculated over the full vertical extent of all calibration quality clouds identified by the V3 1064 nm calibration algorithm. As seen below the solid orange line in Figure 11b, the V3 1064 nm calibration coefficients for October 2010 are biased by the inadvertent inclusion of a non-negligible fraction of water clouds.

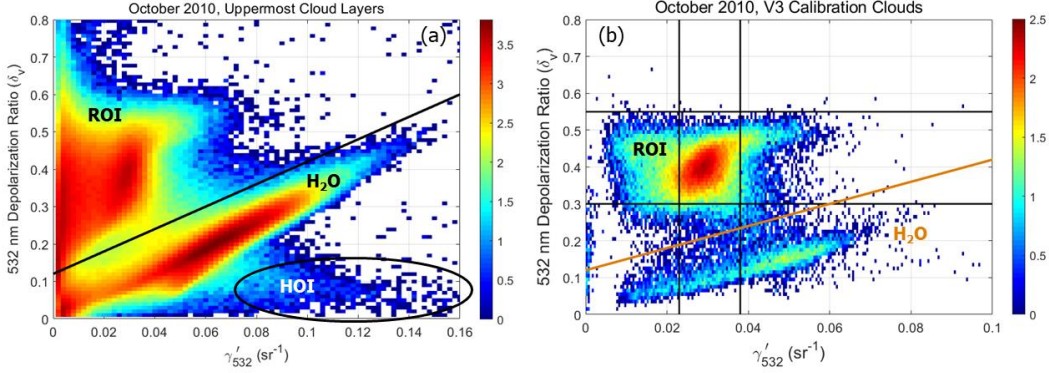

**Figure 11: panel (a) shows the joint occurrence frequency of** $\delta_v$ **and** $\gamma'_{532}$ **for clouds measured by CALIOP during October 2010. Only layers detected at 5 km horizontal resolution that are the uppermost layer in each profile are included. The solid black line**





differentiates randomly oriented ice clouds (above the line) from water clouds (below the line). Clouds containing horizontally oriented ice crystals occur within the oval at the bottom of the plot. Panel (b) shows the joint occurrence frequency of $\delta_v$ and $\gamma'_{532}$ for clouds used in the V3 1064 nm calibration analysis. The population of points below the green threshold line quantifies the occurrence frequency of water clouds in the October 2010 V3 calibration data set. In both plots, the colors indicate $\log_{10}$ of the number of samples in each grid cell.

V3 calibration quality clouds were selected based on a scattering intensity requirement (i.e., the magnitude of the attenuated scattering ratios) designed to reduce bias errors due to molecular contributions to the total scattering from the clouds. In V4 this scattering intensity criterion is satisfied using $\gamma'_{532}$, with the contributions from molecular scattering being accounted for by the $dg_{532}$ term in Eq. (11). An example of the differences in calibration cloud sample sizes associated with these two metrics is illustrated in Figure 12. The V3 calibration analysis identified seven calibration quality clouds in this scene, shown as intermittent occurrences within the brightest white regions circled in blue between 12 km and 14 km and clustered near 1.2° N and 3.4° S. Clearly these V3 calibration quality "clouds" would more accurately be described as "cloud fragments", as those regions for which three contiguous bins of $R'_{532}(z)$ exceed 50 typically represent only a small fraction of the full vertical extent of the cloud identified by the L2 layer detection scheme. Of the 160 detected at 5 km horizontal resolution by the V4 analysis, 116 had 532 nm integrated attenuated backscatters in the acceptable range of 0.023 sr$^{-1}$ < $\gamma'_{532}$ < 0.038 sr$^{-1}$, amounting to a 22-fold increase in the number of potential calibration quality clouds.

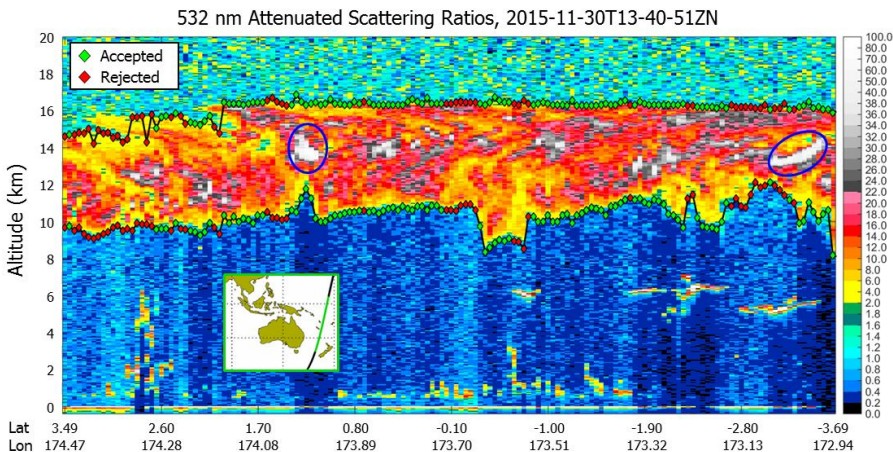

**Figure 12: 532 nm attenuated scattering ratios, averaged to 5-km horizontally and 60 m vertically, for an extended cirrus layer in the southwest Pacific near New Caledonia on 30 November 2015. Potential V3 calibration opportunities ($R'_{532}$ > 50) within the cirrus layer are shown in white. V4 cloud boundaries are indicated by filled diamonds. The boundaries of those clouds for which 0.023 sr$^{-1}$ < $\gamma'_{532}$ < 0.038 sr$^{-1}$ are shown in green. The boundaries of clouds having $\gamma'_{532}$ outside this range are shown in red.**

### 4.1.4 Comprehensive selection strategy implemented in V4

Summarizing the criteria described in the previous subsections, clouds selected for use in the V4 1064 nm calibration algorithm are detected using the same layer detection algorithm that is used in the CALIOP level 2 analyses and are required to meet all of the following specifications.




(a) The cloud must be the uppermost layer detected in a profile averaged to a 5-km horizontal resolution (Sect. 4.1.1).

(b) The boundaries and vertical extent of this uppermost layer are constrained by the local tropopause height at the upper end and the Earth's surface at the lower end (Sect. 4.1.1).

(c) The temperature at the cloud geometric midpoint must be colder than −35° C (Sect. 4.1.2).

(d) The layer-integrated 532 nm volume depolarization ratio must fall within a range of 0.3 to 0.55. The rationale for the lower limit is described in Sect. 4.1.2. The upper limit is defined to eliminate unusually large noise excursions that can occur during daytime measurements of cirrus above bright clouds or desert surfaces, or during both daytime and nighttime when transiting the South Atlantic Anomaly (SAA; see Noel et al., 2014).

(e) The layer-integrated 532 nm attenuated backscatter is restricted to a range of $0.023\ \mathrm{sr}^{-1} < \gamma'_{532} < 0.038\ \mathrm{sr}^{-1}$ (Sect. 4.1.3).

Enforcing these criteria ensures a substantially more homogenous population of clouds than was used in V3. Water clouds are effectively eliminated by the $T_{mid}$ and $\delta_v$ requirements; clouds dominated by horizontally oriented ice crystals are rejected by the $\gamma'_{532}$ and $\delta_v$ limits; and polar stratospheric clouds are excluded by the altitude restrictions. Furthermore, the V4 cloud selection requirements yield a far larger number of calibration quality clouds with a much more uniform distribution as a function of latitude. Figure 13 compares the number of nighttime calibration quality samples obtained during October 2010
for V3 (panel a) to the number of samples that would have been obtained had conditions (a) through (e) above been applied instead (panel b). The V4 selection parameters are seen to provide a much more uniform sampling as a function of latitude, while at the same time delivering a substantially larger number of total samples (59,675 in V3 vs. 92,132 in V4).

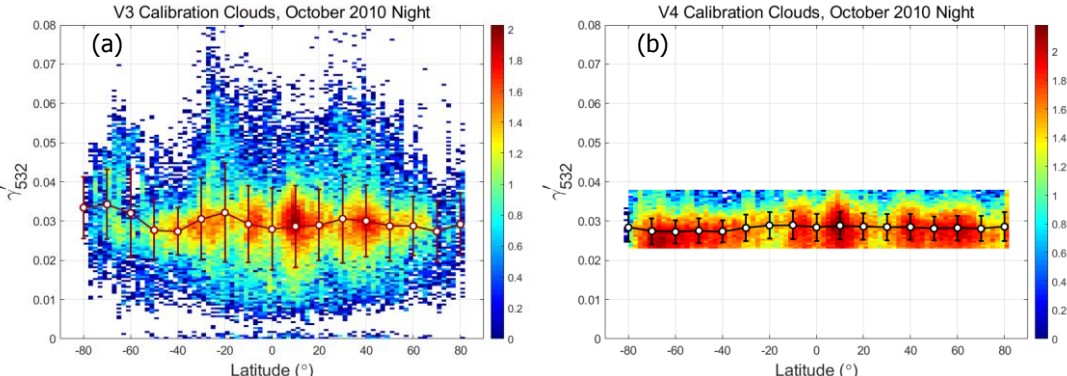

**Figure 13: (a)** $\gamma'_{532}$ **for all V3 calibration clouds as a function of latitude, and (b)** $\gamma'_{532}$ **for all V4 calibration clouds as a function of**
**latitude for October 2010 nighttime measurements. The filled circles in each plot represent** $\gamma'_{532}$ **mean values over 2° latitude increments; error bars indicate ± one standard deviation about the means. The colors indicate** $\log_{10}$ **of the number of samples in each grid cell.**



## 4.2 Characterizing intra-orbit changes using multi-granule data averaging

The primary motivation for the complete redesign of the CALIOP 1064 nm calibration scheme is to accurately characterize the time-varying behavior of the calibration scale factors. As illustrated in Figure 5, these changes occur on multiple time scales, from intra-orbit to seasonal. Designing an effective data averaging scheme thus becomes a question of balancing requirements in two time dimensions: along-track within a single granule, and again across multiple granules. Specifically, we need to accumulate a sample size large enough to minimize the random uncertainty in our estimates of $f_{V4}$, while at the same time (a) limiting the extent of the along-track averaging in order to reliably capture the dependence of scale factors with respect to granule elapsed time, and (b) limiting the duration of our multi-granule averaging window to prevent smearing of legitimate changes in the scale factors that occur on weekly-to-seasonal time scales. The (not to scale) dimensions of the averaging window developed for the V4 1064 nm calibration scheme are illustrated in Figure 14. The red boxes indicate notional averaging regions that extend both along-track (i.e., north–south within any one granule) and across-track (i.e., in the east–west direction, spanning multiple granules).

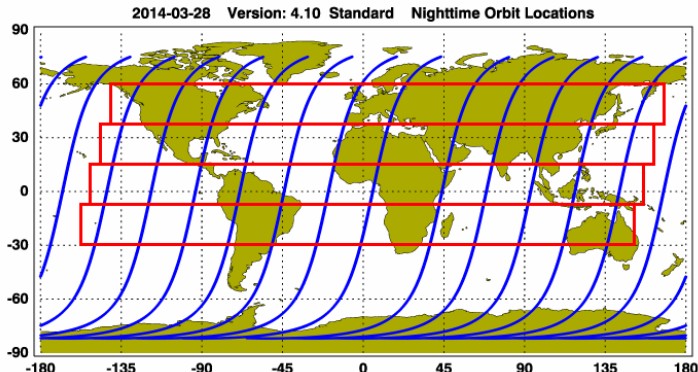

**Figure 14: Nighttime orbit tracks for 28 March 2014 (in blue), overlaid with notional averaging domains (red boxes) that extend over two time dimensions; i.e., traveling along-track (north–south) within individual granules, and spanning the same along-track distance across multiple granules (east–west)**

The driving factor in sizing this two-dimensional averaging window is the number of calibration quality clouds that can be measured in the cloud-sparse seasons and regions of the planet. For the V3 calibration procedure, these regions are indicated by the white grid cells shown in Figure 6. But because V4 uses entirely different cloud selection criteria, the 'cloud sparse' seasons and regions of the planet are also quite different. Figure 15 shows V4 calibration cloud occurrence frequency as a function of granule elapsed time in increments of 90 s (equivalent to an along-track averaging distance of ~ 605 km) for the months of January, April, July, and October 2014. For nighttime data (Figure 15a), granule elapsed time begins at the day-to-night terminator in the northern hemisphere and tracks the temporal progress of the descending node of each orbit. Granule elapsed time for daytime data (Figure 15b) begins in the southern hemisphere and tracks the ascending node of each orbit. For the nighttime data, a minimum value of 276 calibration quality clouds occurs during July at a median granule elapsed time of 1215 seconds (equivalent to ~ 15° S). For the daytime data, a minimum value of 342 calibration quality



clouds occurs during January at a median granule elapsed time of 495 seconds (equivalent to ~ 80° S on the ascending node). Given that the random relative uncertainty in CALIOP's assumed value of $\chi_{cirrus}$ is ± 0.25 (Vaughan et al., 2010), reducing this uncertainty by a factor of 10 requires averaging 100 or more independent samples. In the V4 calibration procedure we achieve this goal at an along-track temporal resolution of 90 s by using a fixed 7-day averaging window, encompassing a

maximum of 105 granules, centered about the current orbit location (i.e., ~ ±54 granules from the current granule). This strategy typically yields well over 250 samples per average, though, as demonstrated in Figure 15, the total for any average varies by both season and location. These averaging intervals are uniformly applied whenever the instrument is in continuous data acquisition mode. As discussed in Getzewich et al. (2018), interruptions (e.g., for periodic boresight alignments, as described in Hunt et al., 2009) require a reboot of the calibration procedures at both wavelengths. When these

reboots occur, the data averaging intervals are reinitiated. For a variety of reasons, the calibration coefficients and scale factors can be notably different immediately before and after an interruption (Getzewich et al., 2018). Section 4.3.2 discusses some consequences of these reboots that are specific to the 1064 nm calibration procedures.

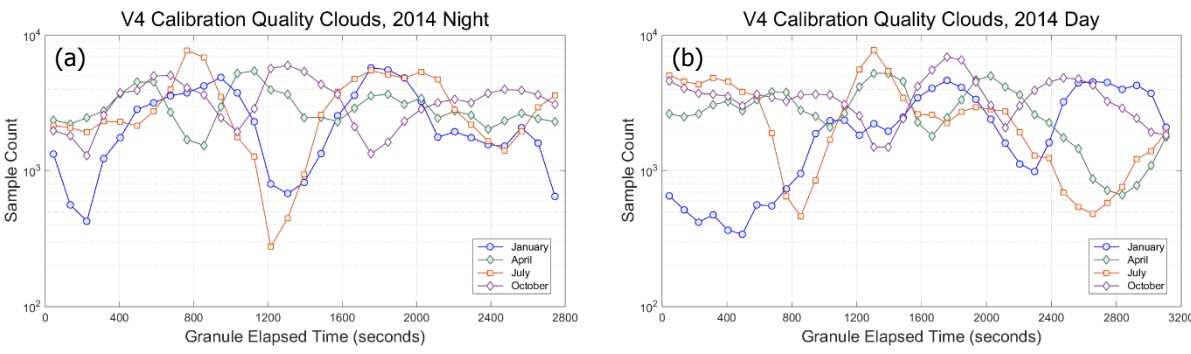

**Figure 15: V4 calibration cloud occurrence frequency as a function of granule elapsed time (90 second bins) for January, April,**
**July, and October 2014. Panels (a) and (b) show, respectively nighttime data and daytime data.**

### 4.3    Uncertainty estimates

The calibration coefficients estimated by CALIOP 1064 nm calibration algorithm are subject to both random uncertainties, which can be substantially reduced by applying the appropriate averaging techniques, and systematic bias error, which cannot be reduced by averaging. The sections below discuss both types of errors and describe how input uncertainties
propagate into the final values of the 1064 nm calibration coefficients.

#### 4.3.1    Random uncertainties

The random uncertainties in the V4 calibration coefficients are derived using the same formalism used in V3, but with $g_\lambda$ replacing $\left\langle X'_\lambda(r) \right\rangle$; i.e.,





$$\left(\frac{\Delta C_{1064}}{C_{1064}}\right)^2 = \left(\frac{\Delta f_{V4}}{f_{V4}}\right)^2 + \left(\frac{\Delta C_{532}}{C_{532}}\right)^2 = \left(\frac{\Delta \chi_{cirrus}}{\chi_{cirrus}}\right)^2 + \left(\frac{\Delta g_{1064}}{g_{1064}}\right)^2 + \left(\frac{\Delta g_{532}}{g_{532}}\right)^2 + \left(\frac{\Delta C_{532}}{C_{532}}\right)^2, \qquad (15)$$

where $\Delta C_{532}$ and $\Delta f_{V4}$ depend critically on the amount of averaging done when deriving the required estimates of $C_{532}$ and $f_{V4}$. Nighttime and daytime derivations for $\Delta C_{532}/C_{532}$ are given in, respectively, Kar et al. (2018) and Getzewich et al. (2018). Random uncertainties for the 532 nm calibration coefficients are typically on the order of 1.5% or less, both at night

and during the day. The multi-granule moving window averaging scheme described in Sect. 4.2 is specifically designed to minimize random uncertainties in $f_{V4}$. Figure 16 provides an example. Figure 16a shows the means and standard deviations for the calibration scale factors acquired over 90 second intervals of granule elapsed time during the 7-day period from 24 November 2015 to 30 November 2015. Figure 16b shows the number of samples acquired in each 90 second time bin. The minimum sample count is 317, occurring at ~81.7° S during the daytime. The relative uncertainties in the mean values of $f_{V4}$

in each 90 s interval $\left(\text{i.e., standard deviation} / \left(\text{mean} \times \sqrt{\text{sample counts}}\right)\right)$ range between 0.11 % and 0.40 % at night (mean = 0.22 % ± 0.07 %) and 0.17 % and 0.52% during the day (mean = 0.29 % ± 0.09 %). Since $\chi_{cirrus}$ is a constant for all calculations, these uncertainties quantify the random variability in the $g_{1064}/g_{532}$ term of $f_{V4}$. But by averaging many samples we also reduce the random uncertainty in our estimate of $\chi_{cirrus}$. In this example, the relative uncertainty attributed to $\chi_{cirrus}$ is reduced from a single sample value of ~ 25 % to mean values of 0.93 % ± 0.20 % during the day and 0.97 % ± 0.24 % at

night. Both in this example and throughout the entire V4 data set, $\chi_{cirrus}$ remains the dominant random uncertainty in estimating $f_{V4}$.

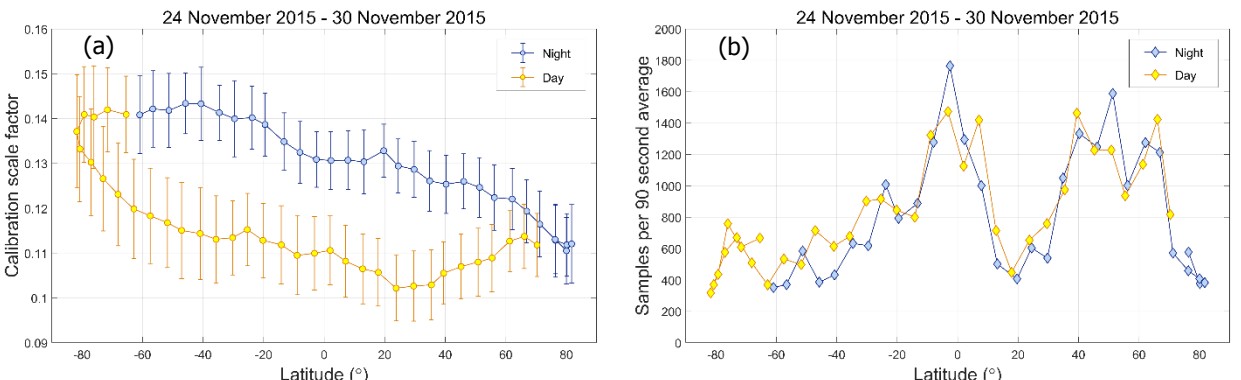

**Figure 16: (a) mean values (filled circles) and single-sample standard deviations (error bars) for the calibration scale factors averaged over 90 second intervals during the 7-day period from 24 November 2015 to 30 November 2015; (b) the number of**
**calibration quality clouds sampled in each 90 second interval.**

The CALIOP V4 data products report estimates of random uncertainties in the 532 nm and 1064 nm calibration coefficients on a profile-by-profile basis. Figure 17a plots the mean values of the relative calibration coefficient uncertainties at both





wavelengths as functions of latitude for all of November 2015. The dip in sample counts shown at ~ 20° N in the right-hand panel of Figure 16 is echoed by the increase in 1064 nm calibration uncertainty seen at the same latitude in Figure 17. The mean and median relative uncertainties for all 1064 nm calibration coefficients computed from 1 January through 31 December, 2015 are shown in Figure 17b and further summarized in Table 2. Taken over the full year and the full globe, the

5 median relative uncertainties during the daytime are 1.77 ± 0.41 %. Nighttime uncertainties are slightly lower, at 1.63 ± 0.29 %. Median uncertainties remain below 2 % daytime and nighttime between ~60° S and ~60° N. The largest relative uncertainties occur in the SAA and for daytime measurements in the polar summers. In polar summers, the daytime 532 nm calibration coefficients and uncertainties cannot be calculated directly, but instead are interpolated between the last known-to-be-valid calibration coefficients in the daytime portion of the orbit and the first last known-to-be-valid calibration

coefficients in the nighttime portion of the same orbit (see Fig. 4 and Sect. 3.7 in Getzewich et al., 2018).

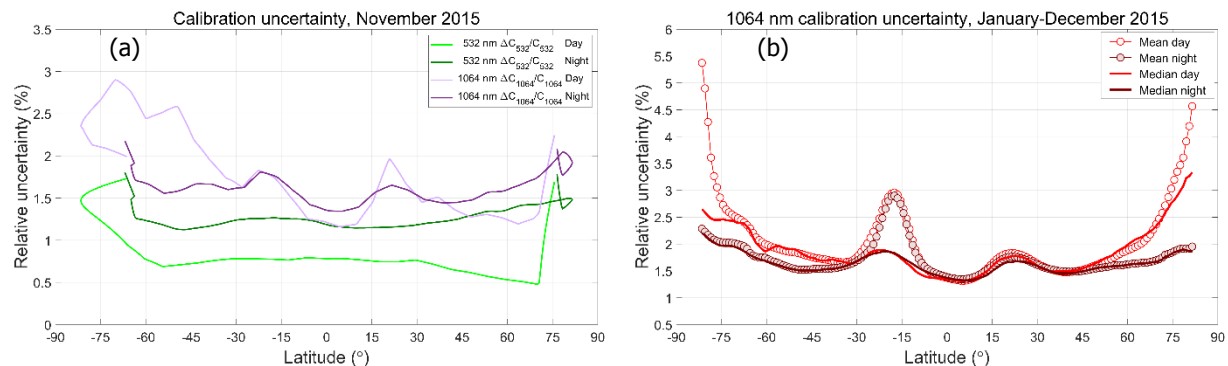

**Figure 17: (a) mean relative calibration coefficient uncertainties, daytime and nighttime, at 532 nm (greens) and 1064 nm (purples) for November 2015; (b) mean and median relative calibration uncertainties at 1064 nm for all data acquired during 2015. In panel (b), the large excursion in the mean uncertainties at ~ 20° S is due to increased uncertainties in the 532 nm calibration coefficients**
**due to high radiation noise in the SAA (Hunt et al., 2009; Noel et al., 2014).**

**Table 2: summary of CALIOP V4 single profile relative calibration coefficient uncertainties for all 1064 nm attenuated backscatter profiles acquired during 2015.**

|  | Day (%) | Night (%) |
|---|---|---|
| Min | 0.02 | 1.24 |
| Max | 42.05 | 25.38 |
| Median | 1.77 | 1.63 |
| MAD | 0.41 | 0.29 |
| Mean | 2.24 | 1.75 |
| Standard deviation | 1.91 | 0.74 |
| 95th percentile | 4.67 | 2.37 |
| Samples | 308,277,495 | 271,947,645 |





### 4.3.2    Bias errors

The V4 CALIOP calibration algorithms are specifically designed to accurately capture small-scale thermal changes that manifest themselves as intra-granule changes in the calibration coefficients at both 532 nm and 1064 nm. However, unexpected changes to CALIOP's on-board thermal environment can introduce bias errors into the 1064 nm scale factor

calculations. Whenever lidar operations are temporarily halted – e.g., due to space weather anomalies or off-nominal instrument behavior – the instrument is commanded to "safe" mode and the standard operating temperatures within the transmitter and receiver are no longer rigorously maintained. When the lidar is subsequently restarted after a long duration outage (e.g., one or more days), 36 to 72 hours of continuous operation can be required before full thermal stability is reestablished. The detector gains for both the PMTs and the APD are temperature sensitive, so during this warm-up period

the calibration coefficients for both channels will approach their steady state behaviors, though not necessarily at the same rate.

The effects of the changing detector gains during the instrument warm-up period are illustrated in Figure 18a, which shows the granule mean of the estimated $\chi'$, $\langle \chi' \rangle$, retrieved for calibration quality clouds measured during all nighttime granules between 11 April 2014 at 17:45:33 UTC (granule number 301) and 30 April 2014 at 13:21:37 UTC (granule number 849).

Due to space weather considerations (i.e., an elevated 10 MeV proton flux), the CALIPSO payload was placed in "safe" mode at 08:29:42 UTC on 19 April 2014, and no instrument data were collected until the payload was restarted at 16:26:07 UTC on 22 April 2014. Prior to shut down, $\langle \chi' \rangle$ oscillates consistently around the expected value of 1.01. However, when the lidar was restarted, the initial values of $\langle \chi' \rangle$ are seen to be substantially lower, though over the course of multiple granules, $\langle \chi' \rangle$ gradually and nonlinearly returns to ~ 1.01. This same behavior is clearly evident in the data acquired

following any shutdown of ~ 12 hours or longer.

The exact mechanisms driving this behavior in the calibration cloud color ratios are not yet fully understood. However, as illustrated in Figure 18b, which shows granule mean $\gamma'_{532}$ for the April 2014 time period, the granule mean $\gamma'_{532}$ for calibration quality clouds is essentially unaffected by the time-varying detector gains. The granule mean $\gamma'_{532}$ prior to the data outage (granules 301–519) is $0.0288 \pm 0.009$ sr$^{-1}$. Following the data outage (granules 619–849), the granule mean $\gamma'_{532}$

is essentially unchanged at $0.0289 \pm 0.008$ sr$^{-1}$. The variability within this time series can be largely attributed to the natural variability of $\gamma'_{532}$ for individual calibration quality clouds. Given that $\gamma'_{532} = g_{532} / C_{532}$ remains essentially constant across the data outage, while $\chi' = \gamma'_{1064} / \gamma'_{532} = g_{1064} / (C_{1064} \times \gamma'_{532})$ varies, current investigations are focused on the 1064 nm channel measurements (i.e., $g_{1064}$) and possible time-varying biases in the calculation of $C_{1064}$.





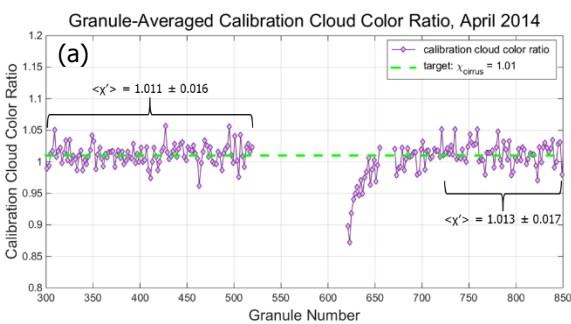 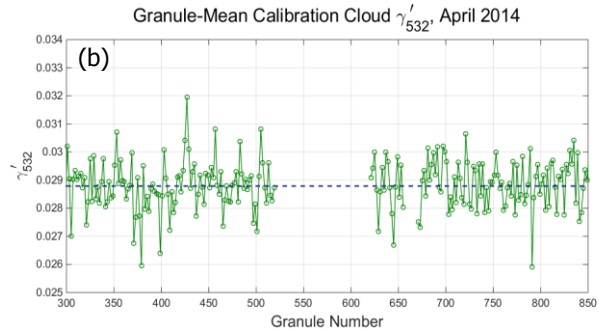

**Figure 18: (a) time series of granule mean $\chi'$ measured for calibration quality clouds detected during nighttime orbit segments beginning on 11 April 2014 (granule 301), and extending through 30 April 2014 (granule 849). Due to adverse space weather, CALIOP was placed in safe mode and thus data is missing for over 3 days, from 19 April 2014 at 08:29:42 UTC to 22 April 2014 at 16:26:07 UTC, spanning granules 521–619. A smaller data gap of just over 8 hours (from 01:09:36 until 09:38:09 UTC on 24 April 2014, spanning granules 657–669) occurs during a satellite drag make-up maneuver. A distinct drop in the magnitude of $\chi'$ occurs when the lidar is restarted on 22 April. Because there were two instrument shutdowns in relatively rapid succession, full recovery to the pre-outage values takes place over ~ 72 hours. (b) granule mean $\gamma'_{532}$ for the same time period. These values remain relatively constant throughout the entire measurement interval, suggesting that the 532 nm calibration appropriately compensates for any time-dependent detector gain changes following an instrument restart.**

Potentially biased estimates of $C_{1064}$ can be identified in the L1 profiles by examining the 'QC_Flag_2' scientific data set (SDS) in the CALIOP level 1b profile products. These QC flags are implemented as 32-bit integers, and interpreted as a series of Boolean values, with each bit indicating a specific warning or error condition. A QC flag #2 of zero indicates that none of these warnings or error conditions has occurred. Those profiles for which $C_{1064}$ may be biased will have bit 27 toggled on (bit 26 if zero-based indexing is used), and thus an otherwise error-free profile with a possibly biased estimate of $C_{1064}$ will have a QC flag #2 of 67108864.

Section 4.3.1 demonstrated that $\chi_{cirrus}$ is the dominant source of random uncertainties in the 1064 nm calibration scale factor error budget. While the random uncertainties in the calibration scale factors due to $\chi_{cirrus}$ can be reduced by averaging, $\chi_{cirrus}$ is also a potential source of irreducible bias errors. The best available estimate of the mean value of $\chi_{cirrus}$ remains 1.01, as determined in V10 and verified by experimentally by Haarig et al., 2016. However, the uncertainty in this estimate is large (± 0.25), and the true value of $\chi_{cirrus}$ may be somewhat different from the value used in the CALIOP V4 calibration algorithm (e.g., 1.00 vs. 1.01, which would introduce a bias of 1 % into the scale factor calculations).

## 5    Performance assessments and comparisons to version 3

The V4 calibration coefficients differ substantially from their V3 predecessors, and these differences manifest themselves on multiple time scales throughout the CALIOP data set. The subsections below compare the calibration coefficients and scale factors generated by the V4 and V3 algorithms, and highlight the V4 improvements in terms of inter-orbit and long-term stability and day-to-night continuity.





## 5.1 Daily-to-monthly changes

The magnitude and spatial variability of the granule-to-granule changes in the calibration coefficients are illustrated in Figure 19, which shows maps of the mean V3 and V4 calibration coefficients for daytime (panels a–c) and nighttime (panels d–f) calculated for March 2015. In the V3 calibration coefficient images (panels a and d), individual granule tracks are

easily discerned, indicating that these granules have unusually large or unusually small calibration coefficients relative to neighboring granules. This "striping" of the V3 1064 nm calibration coefficients occurs because a single mean scale factor is calculated for each granule, and thus when cloud locations or occurrence frequencies shift substantially from one orbit to the next, the concomitant changes in the mean scale factor introduce noticeable granule-to-granule discontinuities in the calibration coefficients. Because the V4 algorithm computes scale factors by averaging over multiple granules,

corresponding to approximately one week of observations, this vertical striping is eliminated in the V4 images and data (panels b and e). Additionally, the influence of the SAA, seen in the nighttime data shown in panels d and f, is now virtually eliminated.

Maps of the monthly mean V3 calibration coefficients divided by the monthly mean V4 calibration coefficients are shown in panels c and f of Figure 19. In this example, the variability between the two data versions extends from –20 % (daytime

southern hemisphere) to +25 % (nighttime northern hemisphere). The changes in the daytime range from +20 % in the northern mid-latitudes to –20 % in Antarctica. Nighttime changes are somewhat more muted in this example, varying between +25 % in the Arctic to –7 % in Antarctica.

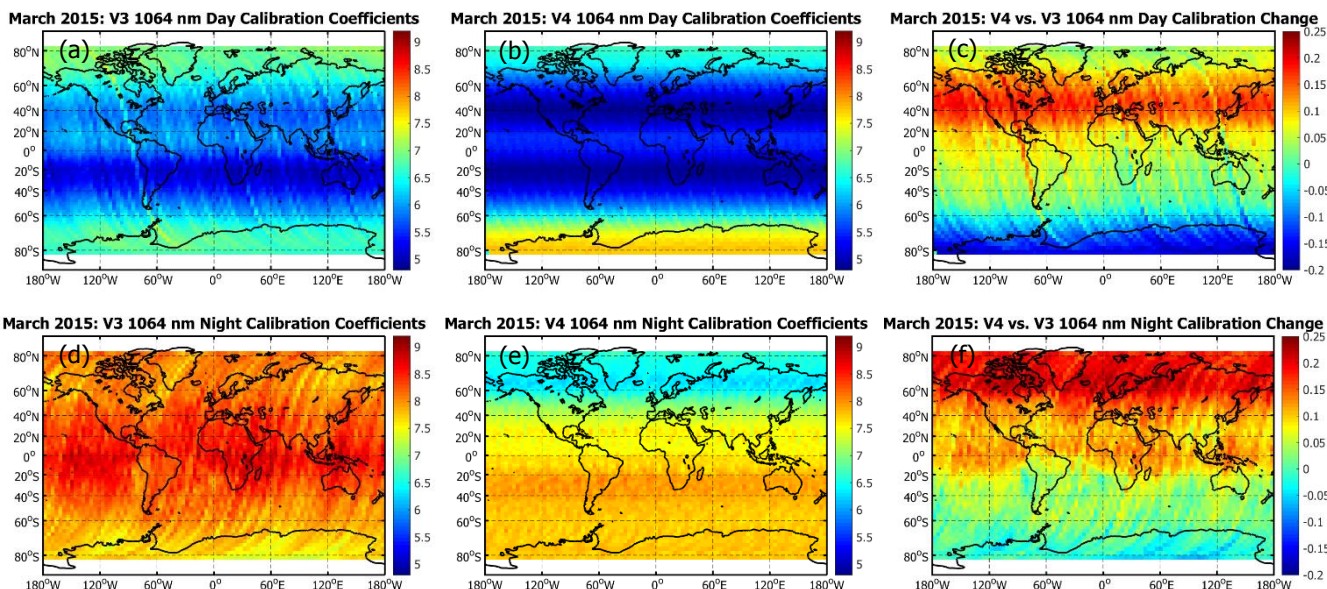

**Figure 19: V3 and V4 calibration coefficients for March 2015. Panels (a) through (c) show daytime mean 1064 nm calibration coefficients (units = km$^{-3}$ sr J$^{-1}$ count); V3 is shown in panel (a), V4 in panel (b), and their ratios (V3/V4) in panel (c). Similarly,**




panels (d) through (f) show nighttime mean 1064 nm calibration coefficients (units = km$^{-3}$ sr J$^{-1}$ count), with V3 is shown in panel (d), V4 in panel (e), and their ratios (V3/V4) in panel (f).

## 5.2    Day-to-night calibration continuity

An important detail that may not be immediately apparent in Figure 19 is shown explicitly in Figure 20, where the March

2015 zonal mean calibration coefficients for both V3 (Figure 20a) and V4 (Figure 20b) are plotted separately for daytime and nighttime granules as a function of latitude. The V3 1064 nm calibration coefficients show large discontinuities when the instrument transitions from day to night (Figure 20a, left side) and again from night to day (Figure 20a, right side). In contrast, the V4 calibration coefficients show no discontinuities crossing the terminators. Because the signals are normalized with respect to electronic gains prior to calibration, this smoothly varying transition across the terminators is the expected

behavior. However, ensuring that the scale factors are continuous across the terminators (e.g., as shown in Figure 16) does not guarantee that the desired outcome actually occurs: the 532 nm calibration coefficients must also be continuous. The substantial changes made in the daytime 532 nm calibration algorithm (Getzewich et al., 2018) are thus an essential precondition for achieving the required continuity at 1064 nm.

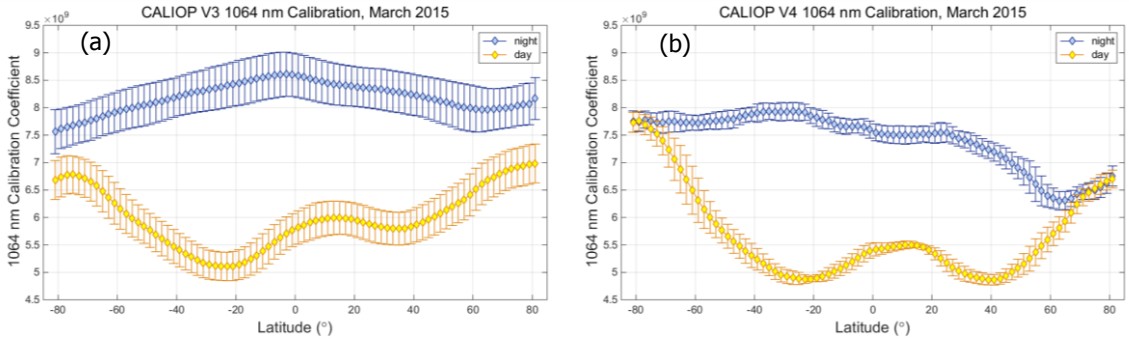

Figure 20: Zonal mean 1064 nm calibration coefficients for March 2015; (a) shows V3 calibration coefficients (units = km$^{-3}$ sr J$^{-1}$),

while (b) shows the V4 coefficients (units = km$^{-3}$ sr J$^{-1}$ count). In both panels, nighttime values are shown in blue and daytime values in yellow. Error bars represent 1 standard deviation about the mean.

## 5.3    Seasonal-to-yearly changes

The seasonal and annual changes between the V3 and V4 nighttime granule-averaged estimates of C$_{1064}$ are illustrated in the

5-year time histories (2013–2017) shown in Figure 21. The V3 calibration coefficients show a strong and persistent seasonal oscillation, with lower values in the boreal summer months and higher values in the boreal winter. Though not eliminated entirely, this oscillatory behavior is markedly reduced in the V4 time history. Figure 22 demonstrates that the magnitude of the V3 oscillations is significantly amplified by the data averaging strategy implemented in V3 calibration procedure. The left panel of this figure shows the daily mean latitude centroid,





$$
C_{\text{latitude}} = \left. \sum_{n=1}^{N} \text{latitude}_n \times f_{V3_n} \middle/ \sum_{n=1}^{N} f_{V3_n} \right. ,
\tag{16}
$$

computed over all nighttime scale factors for each calendar day for which there were CALIOP measurements during 2013–2017. This quantity represents the characteristic latitude associated with the daily mean value of $f_{V3}$. The seasonal oscillations of $C_{\text{latitude}}$ reflect changes in the occurrence frequencies of strongly scattering $\left( \left\langle R'_{532} \right\rangle > 50 \right)$ convective ice

clouds. As seen in Figure 22b, $f_{V3}$ is a decreasing function of latitude, and the scale factors measured in the southern hemisphere are systematically higher than those in the northern hemisphere. The seasonal shifting of $C_{\text{latitude}}$ thus introduces seasonal oscillations in $f_{V3}$ (green line in Figure 22a), which in turn are reflected in the seasonal oscillations seen in the V3 calibration coefficients.

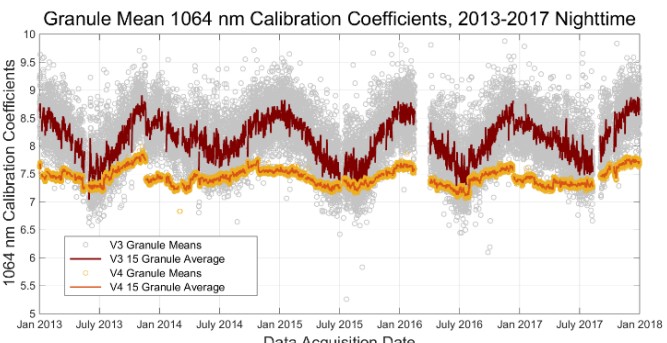

**Figure 21: granule-mean calibration coefficients (scaled by $10^{-9}$, with units = km$^{-3}$ sr J$^{-1}$ count) for V3 and V4 from January 1, 2013 through December 31, 2017. The large data gap from 28 January 2016 through 14 March 2016 is due to a GPS anomaly that interrupted the timekeeping services normally provided by the satellite. Adverse space weather is responsible for the smaller gap from 5 September 2017 through 15 September 2017.**

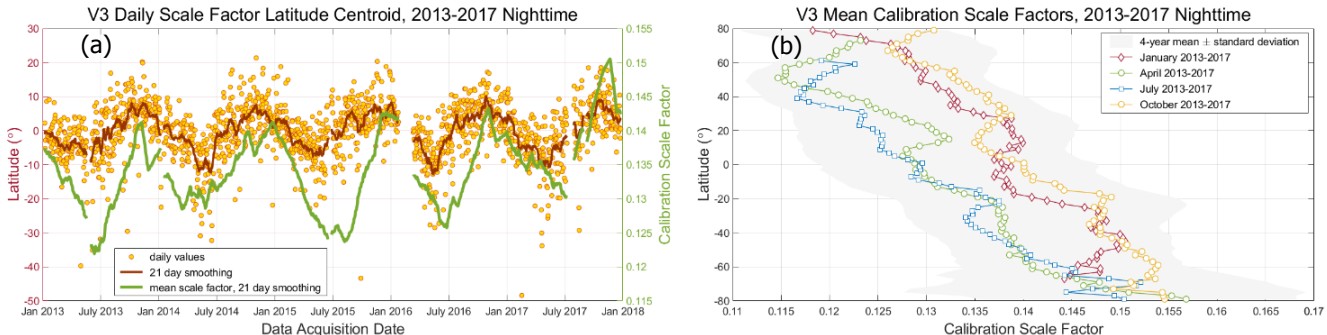

**Figure 22: (a) V3 daily mean latitude centroids (orange circles) measured during 2013–2017. The dark red line shows a 21-day running average. The green line (associated with the right y-axis) shows a 21-day running average of the daily mean V3 calibration scale factors (i.e., as in Eq. (2)). (b) the latitudinal variation of the mean scale factors during the same time period.**



**The gray shaded area represents the 4-year mean ± one standard deviations, with individual lines showing monthly means for January (red diamonds), April (green circles), July (blue squares), and October (orange circles).**

## 6    Comparisons to other techniques and measurements

While CALIOP uses cirrus clouds to calibrate its 1064 nm measurements, other calibration targets are also available.  SNR limitations rule out molecular normalization as an option.  However, water clouds and ocean surfaces offer potentially attractive alternatives; both are typically measured with very high SNR, and their spectral differences in backscatter are well-characterized by theory. In this section we explore the relative merits of using water clouds and/or ocean surfaces as calibration targets.  1064 nm calibration algorithms for both targets are briefly described, and the calibration coefficients derived using these algorithms are compared to the standard values reported in the CALIOP level 1 data products.  In addition, we compare CALIOP's 1064 nm attenuated backscatter profiles to coincident attenuated backscatter profiles acquired independently by the airborne high spectral resolution lidar (HSRL) developed at NASA's Langley Research Center (LaRC).  The results of these studies will allow us to estimate an upper bound on the bias errors in the CALIOP 1064 nm calibration coefficients.

### 6.1    Lidar calibration using ocean surfaces

Ocean surfaces have long been proposed as calibration targets for airborne and space-based lidars (Bufton et al., 1983; Menzies et al., 1998; Josset et al., 2010).  In particular, Menzies et al. (1998) described a technique for using lidar backscatter measurements of the ocean surface to derive estimates of 1064 nm calibration coefficients relative to known 532 nm calibration coefficients.  Leveraging the ocean surface scattering equations in Venkata and Reagan (2016), we develop a Menzies-like approach to obtain estimates of the CALIOP 1064 nm calibration coefficients from the following relationship:

$$C_{1064} = C_{532} \left( \frac{R_{f,532}}{R_{f,1064}} \right) \left( \frac{T^2_{p,532}(0,z_{surface})}{T^2_{p,1064}(0,z_{surface})} \right) \left( \frac{\int_{t_{surface_{top}}}^{t_{surface_{base}}} X'_{1064}(t)\,dt}{\int_{t_{surface_{top}}}^{t_{surface_{base}}} X'_{532}(t)\,dt} \right). \tag{17}$$

In computing these values, the signals are integrated over the time duration of the ocean surface backscatter pulses (i.e., from $t_{surface_{top}}$ to $t_{surface_{base}}$), which are broadened over multiple time intervals (i.e., range bins) by third-order low-pass Bessel filters in the CALIOP receiver electronics (Hu et al., 2007d; Venkata and Reagan, 2016). The $R_f$ terms are the Fresnel reflectance coefficients of seawater, which we take to be 0.0213 at 532 nm and 0.0202 at 1064 nm (Quan and Fry, 1995), and the $T^2$ terms represent the two-way attenuation of the signal due to clouds and/or aerosols between the lidar and the ocean surface. By calibrating relative to the 532 nm channel, we eliminate the need for accurate estimates of wind speeds, wave slope





variances, and whitecap frequencies that would otherwise be required to directly calibrate the 1064 nm channel using ocean surface measurements (Lancaster et al., 2005).

Equation (17) has the same general form as Eq. (1): that is, the 1064 nm calibration coefficient is obtained by multiplying a previously derived 532 nm calibration coefficient by a (possibly time-varying) scale factor computed based on the

differences in backscatter signal magnitudes from some well-characterized target. A one-month comparison of ocean surface scale factors to the cirrus cloud scale factors used to calibrate the V4 data products is shown in Figure 23. The ocean data are derived for daytime measurements during the month of October 2010 between 60° N and 60° S. The latitude limits were enforced to minimize possible sea ice contamination of the ocean surface samples. To further reduce the possible inclusion of sea ice samples, the ocean surface depolarization ratios were constrained to lie between 0 and 0.15 (Lu et al.,

2017). Ocean surface scale factors were computed at single-shot resolution using V4 level 1 profiles in which no clouds were detected in any of the CALIOP level 2 data products. Aerosol loading was minimized by requiring the column integrated attenuated backscatters at 532 nm to lie between 0.0036 sr$^{-1}$ and 0.0176 sr$^{-1}$. Estimates of the aerosol two-way transmittance ratio (i.e., $T^2_{532}/T^2_{1064}$) were obtained from collocated Moderate Resolution Imaging Spectroradiometer (MODIS) optical depth retrievals available in the CALTRACK data products distributed by the AERIS/ICARE Data and

Services Center. 532 nm optical depths were interpolated from the MODIS aerosol optical depths reported at 470 nm and 550 nm. Similarly, 1064 nm optical depths were interpolated from MODIS retrievals at 860 nm and 1240 nm. Only those MODIS retrievals for which the QC flags were greater than zero were used in the calculations.

As seen in Figure 23, the agreement between the ocean surface and cirrus cloud scale factors is reasonably good, with the maximum difference between the median values at any latitude being less than ± 5 %. The global scale factor ratio (ocean

surface medians divided by cirrus cloud medians) over all latitudes is 1.008 ± 0.023. Extending the calculations to include multiple months in different seasons and years (i.e., April 2012, July 2014, and January 2016) yields a global scale factor ratio of 1.021 ± 0.003. While this degree of correspondence between the two techniques is highly encouraging, practical implementation of the ocean surface calibration method is limited to daytime measurements, when independent estimates of aerosol two-way transmittance ratios are available from MODIS.





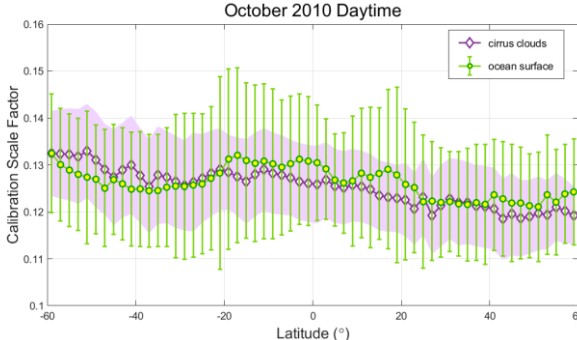

**Figure 23: median 1064 nm calibration scale factors computed using cirrus clouds (purple diamonds) and ocean surface measurements (green circles) for daytime measurements acquired between 60° S and 60° N during October 2010. The pale purple filled region indicates ±1 median absolute distance (MAD) about the median cirrus cloud scale factors. Likewise, the green bars indicate ±1 MAD about the median ocean surface scale factors.**

## 6.2    Lidar calibration using opaque water clouds

Lidar calibration using opaque water clouds was initially proposed by O'Connor et al. (2004), who note that the lidar ratio (i.e., the extinction-to-backscatter ratio) for these clouds, $S_c$, is essentially constant over a large range of droplet size distributions (e.g., $S_c = 18.8 \pm 0.8$ sr at 905 nm), and thus calibration coefficients at a given wavelength can be derived using Platt's equation (Platt, 1973; Hu et al., 2007a); i.e.,

$$\gamma'_c = \int_{z_{top}}^{z_{base}} \beta'(r)\,dr = T_p^{2\eta_p}\left(0, r(z_{top})\right)\left(\frac{1 - T_c^{2\eta_c}\left(r(z_{top}), r(z_{base})\right)}{2\eta_c\,S_c}\right). \qquad (18)$$

Here $\gamma'_c$ is the integrated attenuated backscatter from cloud top to the apparent cloud base, $\eta_c$ is the layer-effective multiple scattering factor within the cloud, and $T_p^{2\eta_p}\left(0, r(z_{top})\right)$ is the particulate two-way transmittance between the lidar and cloud top (or cloud base for an up-looking lidar). Because the clouds are opaque, the two-way transmittance through the cloud is a known value (i.e., $T_c^{2\eta_c} = 0$). Similarly, $S_c$ is also assumed to be known within small and well-defined error bounds. In demonstrating their technique, O'Connor et al. use the multiple scattering model described in Eloranta (1998) to calculate the required estimates of $\eta_c$. Assuming that in-cloud contributions from molecular backscattering can be neglected (a reasonable assumption at 1064 nm), integrating the range-corrected, uncalibrated signal from cloud top to the apparent cloud base yields

$g_c = \int_{z_{top}}^{z_{base}} X'(r)\,dr = C\gamma'_c$, and thus the calibration coefficient is

$$C = \frac{2\eta_c\,S_c\,g_c}{T_p^{2\eta_p}\left(0, r(z_{top})\right)}. \qquad (19)$$





The dominant source of uncertainty in Eq. (19) is $\eta_c$, with uncertainties in the particulate attenuation between the lidar and cloud top being secondary. Using model calculations to estimate $\eta_c$ requires a priori knowledge of the droplet size distributions within the water clouds being used as calibration targets. Presumably CALIOP could obtain droplet size information from collocated MODIS retrievals (e.g., as in Figure 26a). However, validation studies indicate that the MODIS

effective radius estimates can be biased high (Painemal and Zuidema, 2011; Min et al., 2012), and thus the resulting estimates of $\eta_c$ would likewise be biased. (Larger droplet sizes would generate increased multiple scattering in the model, leading to an underestimate of both the true $\eta_c$ and the derived calibration coefficients.)

Wu et al. (2011) used measurements from a multi-wavelength (355 nm, 532 nm, and 1064 nm), zenith-pointing, ground-based lidar to compare 1064 nm calibration coefficients calculated using O'Connor's water cloud method to those derived

using the ice cloud technique with the assumption that $\chi_{cirrus} = 1$. Estimates of $\eta_c$ required for the water cloud retrieval were obtained by applying the Eloranta multiple scattering model to cloud droplet size distributions reported in the MODIS data products. The relative difference between these two calibration data sets was typically less than 15%, and these differences fell within the uncertainty bounds estimated using standard propagation of errors analyses (Wu et al., 2011). When differences of 4% or more were found (5 of 7 comparisons), the water cloud calibration coefficients were uniformly lower,

which may indicate an overestimate of droplet sizes in the MODIS data. In general, the Wu et al. ice cloud calibration coefficients were slightly larger, less variable, and more temporally stable than those obtained using the water cloud technique.

### 6.3   CALIOP calibration using opaque water clouds

For CALIOP and other lidars that directly measure linear depolarization, multiple scattering models are not required. For

these systems, accurate estimates of $\eta_c$ for opaque water clouds can be derived from layer-integrated volume depolarization measurements, $\delta_v$, using

$$\eta_{c,\lambda} = \left( \frac{1 - \delta_{v,\lambda}}{1 + \delta_{v,\lambda}} \right)^2 \tag{20}$$

(Hu et al., 2007b; Roy and Cao, 2010). However, while CALIOP makes dual-polarization measurements at 532 nm, the 1064 nm channel measures only the total backscattered energy, and not the separate parallel and perpendicular components.

To obtain estimates of the 1064 nm multiple scattering factors, we employ an empirical relationship developed by Hu et al. (2007c) that expresses the mean extinction coefficient and effective droplet radius at the top of opaque water clouds as a function of layer-integrated volume depolarization ratio; i.e.,

$$\sigma_{wc}\, \mathcal{X}_{wc}^{-1/3} = 1 + 135 \left( \frac{\delta_v}{1 + \delta_v} \right)^2 , \tag{21}$$





where $\sigma_{wc}$ is the volume extinction coefficient at cloud top and $\delta_v$ is obtained as in Eq. (13). $\mathcal{X}_{wc}$ is the Mie scattering size parameter of the mean droplet radius at cloud top; i.e., $\mathcal{X}_{wc} = 2\,\pi\,R_{wc}\,/\,\lambda$, where $R_{wc}$ is the mean droplet radius at cloud top and $\lambda$ is the measurement wavelength. This relationship was derived from extensive Monte Carlo simulations of Mie scattering in opaque water clouds and was developed specifically for the analysis of CALIOP 532 nm daytime data, for

which direct measurements $\delta_v$ are readily available (Hu et al., 2007c).

Repurposing Eq. (21) to retrieve estimates of CALIOP $\delta_v$ at 1064 requires

(a) recognizing that the Mie scattering size parameter for 1064 nm measurements is half of the 532 nm value, so that $\mathcal{X}_{wc,1064} = \frac{1}{2} \times \mathcal{X}_{wc,532}$; and

(b) assuming the Mie scattering extinction efficiencies at 532 nm and 1064 nm are both 2. As this assumption is very well

founded for particle radii greater than $\sim 3\ \mu m$, we further assume that $\sigma_{wc,1064} = \sigma_{wc,532}$.

Adopting these assumptions, defining

$$D_{\lambda} = \delta_{v,\lambda} \Big/ 1 + \delta_{v,\lambda}\ ,$$  (22)

and dividing Eq. (21) evaluated at 1064 nm by Eq. (21) evaluated at 532 nm yields

$$D_{1064} = \frac{\delta_{v,1064}}{1+\delta_{v,1064}} = \sqrt{\frac{2^{\frac{1}{3}}\left(1+135\cdot D_{532}{}^{2}\right)-1}{135}}\ ,$$  (23)

from which we subsequently derive

$$\eta_{c,1064} = \left(\frac{1-\delta_{v,1064}}{1+\delta_{v,1064}}\right)^{2} = \left(1-2\,D_{1064}\right)^{2}.$$  (24)

This transform produces 1064 nm multiple scattering factors that, on average, are smaller than those at 532 nm by $\sim 19\ \%$ (i.e., $\eta_{532}\,/\,\eta_{1064} = 1.190 \pm 0.046$), indicating that the multiple scattering contributions to the total 1064 nm backscatter signals are relatively higher than the contributions at 532 nm.

Substituting Eq. (24) into Eq. (19) and evaluating at 1064 nm yields





$$C_{1064} = \frac{2\left(1-2D_{1064}\right)^2 S_{1064}\, \mathcal{g}_{1064}}{T_{p,1064}^{2\eta_{p,1064}}\left(0, r\left(z_{top}\right)\right)}. \tag{25}$$

Dividing both sides of Eq. (25) by the 1064 nm calibration coefficients computed in the V4 level 1 data products allows us to replace $\mathcal{g}_{1064}$ with $\gamma'_{1064}$, and thus readily evaluate the right-hand side using the layer properties reported in the CALIOP V4 L2 5 km cloud layer products (Vaughan et al., 2018); i.e.,

$$\frac{C_{1064,water}}{C_{1064,cirrus}} = \frac{2\left(1-2D_{1064}\right)^2 S_{1064}\, \gamma'_{1064}}{T_{p,1064}^{2\eta_{p,1064}}\left(0, r\left(z_{top}\right)\right)} = \frac{2\eta_{c,1064}\, S_{1064}\, \gamma'_{1064}}{T_{p,1064}^{2\eta_{p,1064}}\left(0, r\left(z_{top}\right)\right)}. \tag{26}$$

After the division, the left-hand side becomes the relative calibration coefficient; i.e., the ratio between the calibration coefficients computed using water clouds (in the numerator) and cirrus clouds (the denominator).

Using $S_{1064} = 18.2$ sr (Pinnick et al., 1983), we derived the water cloud-to-ice cloud relative calibration coefficients for all of 2015 using the following data selection criteria. The analysis was first limited to opaque water clouds detected over oceans
at CALIOP's standard 5 km horizontal averaging resolution. We further required that

a)    only one layer was detected in each 5 km column, and that same layer was also detected in each single shot profile that comprised the 5 km average (i.e., the layer was spatially homogeneous and robust throughout);

b)    the 532 nm integrated attenuated backscatter above cloud top was within ± 2 standard deviations of the value that would be expected for pristine air; given that the mean cloud top is above 1.5 km (i.e., comfortably above a typical marine
boundary layer height of 0.5 km), enforcing this condition allows us to assume that $T_{p,1064}^{2\eta_{p,1064}}\left(0, z_{top}\right) \approx 1$ (but note: due to very large differences in background noise, the magnitudes of the standard deviations differ markedly for nighttime and daytime measurements); and

c)    the 5 km layer cloud-aerosol discrimination (CAD) score was greater than 90 and less than or equal to 100 (Liu et al., 2018) and the ice-water phase confidence assessment was "high" (Avery et al., 2018), thus guaranteeing the highest
possible classification confidence in both feature type and cloud phase.

To further ensure that only liquid water clouds were included, the mid-layer temperature was required to be above 0 °C. Potential outliers were eliminated by removing all candidate clouds for which either the integrated attenuated backscatter at 532 nm or 1064 nm or the 532 nm integrated volume depolarization ratio fell outside the limits established by the median population values plus or minus two median absolute deviations (MAD). This final filtering step reduced the total number of
layers considered by ~ 14 %.





Distributions of these water-to-ice relative calibration coefficients are shown in Figure 24 and summarized in Table 3. Considered globally, the daytime 1064 nm water cloud calibration shows essentially no bias: the mean 1064 nm relative calibration is $1.004 \pm 0.121$. But the nighttime results are not so reassuring, as the nighttime mean 1064 nm relative calibration is $1.073 \pm 0.112$. The same pattern of day-night differences is seen when water clouds are used to compute

relative calibration coefficients at 532 nm. Using $S_{532} = 18.6$ sr (O'Connor et al., 2004), the daytime mean shows a relatively small bias, at $1.017 \pm 0.116$, that is consistent with the daytime 532 nm attenuated backscatter coefficient biases (1.0 % ± 3.5 %) established via validation studies using LaRC HSRL measurements (Getzewich et al., 2018). However, the nighttime mean ($1.126 \pm 0.095$) shows a bias that is approximately seven times higher than in the daytime (0.126 vs. 0.017) and is substantially larger than the bias that would be expected based on HSRL validation studies (1.6 % ± 2.4 %; see Kar et al.,

2018). While the exact cause for these day-night differences has not yet been definitively ascertained, at present the most likely culprit is thought to be the non-ideal detector response at 532 nm (Hunt et al., 2009). The behavior of the 532 nm photomultipliers is well-described by McGill et al., 2007: "Following a strong impulse signal, such as from the Earth's surface or a dense cloud, the signal initially falls off as expected but at some point begins decaying at a slower rate that is approximately exponential with respect to time (distance)." This exponential decay artificially broadens the vertical extent

of dense water clouds measured at 532 nm.

The design of the CALIOP layer detection scheme also contributes to the biases introduced into the data products by this non-ideal response. For any layer detected, the initial estimate of layer base is continually lowered so long as the slope of the backscatter signal remains negative with respect to range from the lidar (Vaughan et al., 2009). The high background noise characteristic of daytime measurements of dense (and very bright) water clouds largely prevents excessive lowering of

layer base altitudes. However, no such inhibitions are present in the nighttime data, and as a result the opaque water clouds in this study show large day vs. night differences in geometric thickness (medians of $0.509 \pm 0.184$ km daytime vs. $0.838 \pm 0.174$ km nighttime), $\gamma'_{532}$ (day-night median values of 0.0636 sr$^{-1}$ and 0.0761 sr$^{-1}$, respectively), $\gamma'_{1064}$ (day-night medians of 0.0761 sr$^{-1}$ and 0.0893 sr$^{-1}$), and $\delta_v$ (day-night medians of 0.2098 and 0.2287). While the 1064 nm detector does not exhibit the same non-ideal response shown by the 532 nm detectors, the day vs. night differences in $\gamma'_{1064}$ appear because at

the 5 km horizontal averaging resolution the CALIOP layer detection algorithm searches only the 532 nm measurements to establish layer base and top altitudes. On-going algorithm and data product development is expected to minimize artifacts introduced by non-ideal detector response in future data releases.





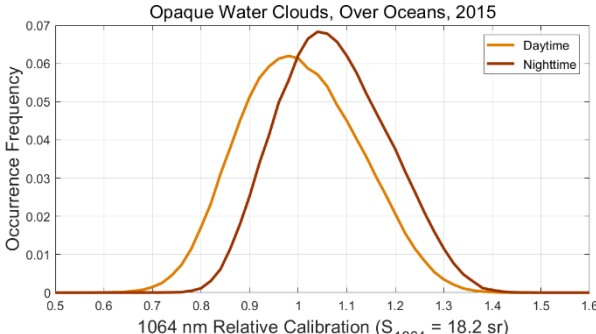

**Figure 24: occurrence frequencies of the 1064 nm relative calibration coefficients (water cloud technique / ice cloud technique) for opaque water clouds detected over global oceans during all of 2015. Daytime results are shown in orange and nighttime results in dark red.**

**Table 3: descriptive statistics for the relative calibration coefficients computed at both wavelengths for opaque water clouds detected over global oceans during all of 2015; values at 1064 nm report water cloud calibration coefficients divided by cirrus cloud calibration coefficients, whereas values at 532 nm report water cloud calibration coefficients divided by the 532 nm calibration coefficients computed using molecular normalization (Kar et al., 2018; Getzewich et al., 2018).**

|  | 532 day | 532 night | 1064 day (Eq. (26)) | 1064 night (Eq. (26)) |
|---|---|---|---|---|
| Min | 0.445 | 0.597 | 0.511 | 0.511 |
| Max | 1.520 | 1.514 | 1.547 | 1.526 |
| Median | 1.009 | 1.116 | 0.999 | 1.067 |
| MAD | 0.092 | 0.075 | 0.099 | 0.091 |
| Mean | 1.017 | 1.126 | 1.004 | 1.073 |
| St. Dev. | 0.116 | 0.095 | 0.121 | 0.112 |
| Samples | 406,058 | 234,441 | 406,058 | 234,441 |

### 6.4    Color ratios of opaque water clouds

Accurate estimates of layer-integrated attenuated color ratios (i.e., $\chi'_{layer} = \gamma'_{1064} / \gamma'_{532}$) are critically important for the CALIOP cloud-aerosol discrimination algorithm, which relies on spectral differences in attenuated backscatter as one of the fundamental measurements that can reliably distinguish clouds from aerosols (Liu et al., 2018; Zeng et al., 2018). Furthermore, having accurate estimates of $\chi'_{layer}$ for opaque water clouds now enables reliable retrievals of Ångström exponents and 1064 nm lidar ratios for aerosol layers lying above these clouds (Chand et al., 2008; Vaughan et al., 2015),

and thus provides opportunities to mine new types of information from the CALIOP measurements. As would be expected, the large changes made in the V4 1064 nm calibration coefficients have had a pronounced effect on both the magnitude and the consistency of the $\chi'_{layer}$ values reported in the CALIOP L2 data products. This is illustrated in Figure 25, which shows zonal mean $\chi'_{layer}$ for opaque water clouds measured both daytime and nighttime in V3 and V4. These clouds are all detected using 5 km horizontal averaging. To minimize the influence of overlying cloud and/or aerosol layers on the $\chi'_{layer}$





measurements, these layers are also the only layers detected within each column. The nighttime V3 data and the daytime V3 data south of 45° N show distinct negative slopes as a function of latitude, with values in the southern hemisphere being 15 – 25 % larger than those in the northern hemisphere. For V3, $\chi'_{layer}$ is reasonably consistent (albeit not constant) for both daytime and nighttime between 45° S and 45° N, but poleward of those latitudes the day and night values diverge

substantially. On the other hand, the V4 measurements remain reasonably constant as a function of latitude during both daytime and nighttime, although there is a consistent difference in magnitude of ~ 3 % on average (daytime higher).

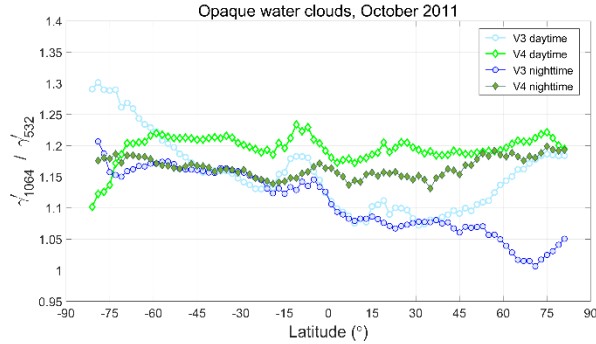

**Figure 25: layer-integrated attenuated color ratio ($\chi'_{layer} = \gamma'_{1064} / \gamma'_{532}$) for opaque water clouds measured during October 2011. V3 values are plotted using blue circles, while V4 values are shown by green diamonds.**

To assess the expected latitudinal changes of $\chi'_{layer}$ we appeal to a combination of Mie scattering calculations and the droplet size distributions for oceanic water clouds derived from MODIS measurements. Figure 26a shows the zonal mean effective radii retrieved from daytime MODIS measurements of opaque water clouds along the CALIPSO orbit track during October 2011. (Because the effective radius retrieval requires input from the visible channels, MODIS does not retrieve effective radius for nighttime measurements.) The MODIS estimates of effective radii are seen to vary as a function of latitude, with

values of 9–14 μm poleward of ~ 45° N and ~ 55° S, mid-range values of 14–18 μm at mid-latitudes, and maximum values of up to ~ 20 μm in the tropics. Figure 26b shows Mie calculations (Liu et al., 2015) of the particulate backscatter color ratio (1064 nm / 532 nm) derived for the size distributions measured in-situ for various marine water clouds (Miles et al., 2000). These backscatter color ratios remain remarkably stable, with a mean value of 1.034 ± 0.034 for mean droplet radii ranging between ~ 4.5 μm and ~ 13 μm, which is similar to size range occupied by the MODIS-derived effective radii poleward of ~

45° N and ~ 55° S (Figure 26a). Consequently, despite the possible high bias in the MODIS effective radii (15–20 % according to Painemal and Zuidema (2011)), we should expect that $\chi'_{layer}$ would likewise remain similarly stable in this region. However, with the possible exception of the nighttime data poleward of ~ 55° S, the CALIOP V3 $\chi'_{layer}$ values shown in Figure 25 are not constant, but instead change substantially as a function of latitude, and diverge in different directions for daytime and nighttime measurements.





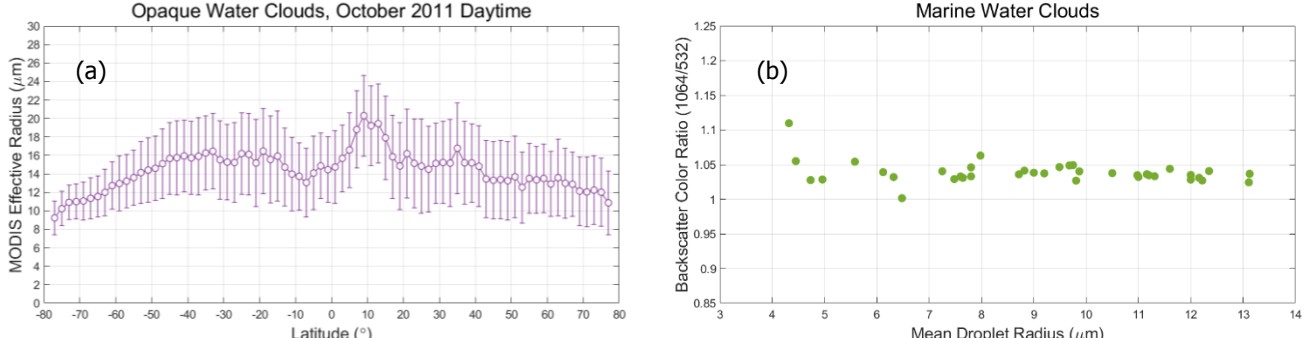

**Figure 26: (a) MODIS retrievals of effective radius for opaque water clouds measured along the CALIPSO orbit track during daytime; (b) Mie calculations of backscatter color ratio (1064 nm / 532 nm) computed for in-situ measurements of marine water clouds as a function of cloud mean droplet radius.**

Due to multiple scattering differences at 532 nm and 1064 nm, the magnitudes of the layer-integrated attenuated color ratios are not expected to be identical to the magnitudes of the unattenuated particulate backscatter color ratios. By using Platt's equation together with the relationship between multiple scattering factors at 532 nm and 1064 nm established in Sect. 6.2.1 and our previous assumptions on $S_{532}$ and $S_{1064}$, we obtain an expected value of $\chi'_{layer}$ in the neighborhood of

$$\left(\eta_{532} \times S_{532}\right)\Big/\left(\eta_{1064} \times S_{1064}\right) \approx 1.19 \times \left(\frac{18.6}{18.2}\right) = 1.216 .$$ This value is very close to the V4 measured daytime global mean of

1.200 ± 0.075, and only slightly higher (~ 4 %) than the V4 nighttime global mean of 1.169 ± 0.062. Furthermore, assuming the extinction coefficients are identical at 532 nm and 1064 nm (i.e., as in Sect. 6.3), the ratio of assumed lidar ratios (18.6/18.2 = 1.022) is entirely consistent with the ratio of particulate backscatter coefficients retrieved from earlier Mie calculations (1.034 ± 0.034). Consequently, our expectation is that, poleward of ~ 45° N and ~ 55° S, the latitudinal variations of $\chi'_{layer}$ and the ratio of particulate backscatter coefficients should be largely identical. And while this holds true for the V4 $\chi'_{layer}$ data, it is decidedly not so for the V3 $\chi'_{layer}$ data.

### 6.5 Availability of 1064 nm calibration targets

The foregoing sections suggest that, at least for daytime measurements, CALIOP has three reasonable choices for a 1064 nm calibration target: ice clouds, water clouds, and ocean surfaces. This appearance of choice, however, is largely illusory when the availability of the three different targets is considered. Figure 27 plots the zonal frequencies for which ice clouds (blue circles), water clouds (green diamonds), ocean surfaces (magenta squares), and aerosol layers (orange line) occur as the uppermost layer detected in a CALIOP 5 km averaged profile as a function of latitude of all data acquired during 2015. Ice clouds are the uppermost layer detected in 38.2 % of the global measurements. Perhaps surprisingly, aerosols have the next highest frequency of occurrence, and are the uppermost layer detected 30.3 % of the time, while the occurrence frequencies for water clouds and liquid ocean surfaces are 18.9 % and 1.1 %, respectively. Sea ice (1 %), land surfaces (6.3 %), and clouds of unknown phase (4.3 %) account for the remainder of the cases. Fully capturing the intra-orbit variability




demonstrated by the scale factors and 1064 nm calibration coefficients (e.g., as seen in Figure 5 and Figure 20) strongly argues for the use of the most frequently observed calibration target. Ice clouds are the clear choice in this regard, as they are detected as the uppermost layer twice as often as water clouds and ~ 35 times more often than the ocean surface.

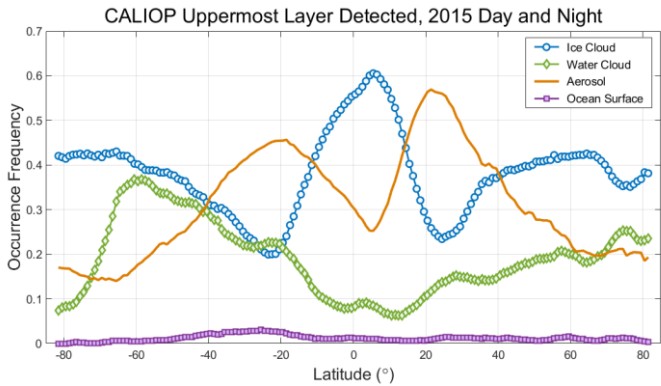

**Figure 27: occurrence frequencies for the uppermost feature detected at 5 km horizontal resolution as a function of latitude for all CALIOP measurements acquired during 2015**

### 6.6    Comparisons to LaRC HSRL measurements

Since launch, the CALIPSO project has partnered with the LaRC HSRL team to conduct a mission-long series of validation under-flights specifically designed to assess CALIOP's calibration accuracy and monitor long-term trends (Rogers et al., 2011). The airborne HSRL's high SNR, downlooking viewing geometry, and ability to measure the same along-track vertical swath as CALIOP yield highly reliable validation measurements in which systematic errors due to aerosol variability are largely eliminated (Gimmestad et al., 2017).

At 532 nm, the HSRL systems use a set of internal calibration procedures to accurately characterize the filter transmittances and detector gain ratios necessary to retrieve aerosol backscatter and extinction coefficients from multiple data channels (Hair et al., 2008). These data have been used extensively to validate the CALIOP 532 nm attenuated backscatter measurements (Rogers et al., 2011 Kar et al., 2018; Getzewich et al., 2018). Here we present an initial comparison of the CALIOP and HSRL attenuated backscatter measurements at 1064 nm. While the LaRC HSRL systems acquire high spectral resolution data at 532 nm, and at 355 nm on HSRL-2 (Burton et al., 2018), at 1064 nm they make elastic backscatter measurements that are calibrated using a variant of the molecular normalization technique. Because the ratio of the calibrated 532 nm HSRL signals provides a direct measurement of the aerosol scattering ratio at 532 nm, regions of minimum aerosol loading are readily identified in the 532 nm profiles. The aerosol scattering ratios at 1064 nm in these minimum loading regions are then estimated using an assumed aerosol backscatter color ratio of $\beta_a(1064) / \beta_a(532) = 0.4$. 1064 nm calibration coefficients are subsequently derived by normalizing the measured 1064 nm signals in the minimum





loading regions to a molecular model that incorporates contributions from the estimated 1064 nm aerosol scattering ratios (Hair et al., 2008).

Figure 28 shows some representative comparisons of collocated 1064 nm attenuated backscatter profiles measured by HSRL (in black) and CALIOP (in red). The left and right panels show daytime data, while the middle panel shows nighttime data.

5 All three cases show regions of relatively clear air above aerosol layers of varying backscatter intensity. In the right-hand panel, signals from dense water clouds are seen beginning at just above 1 km. Visually, the agreement in the "clear air" portions of the profiles appears quite close, whereas the differences in the aerosol-laden regions are generally more pronounced, especially for the daytime cases. The SNR differences between the two sets of measurements are immediately apparent and quite large: the CALIOP data are much noisier than the HSRL data. In part, this difference is due to the

10 amount of data averaged. For the validation flights, HSRL flies directly along the CALIPSO ground track, so the data averaged for each profile covers the same along-track distance. However, the data acquisition times are drastically different. When flying aboard the NASA Langley B-200 aircraft, the typical ground speed for the airborne HSRL is ~ 110 m/s (Rogers et al., 2011). Because CALIPSO moves at ~ 7500 m/s, in just under 53 seconds CALIOP travels the same along-track distance that HSRL would cover in 1 hour. Because the meteorology can change rapidly, these differences in data

15 acquisition times can translate directly into differences in the composition of the scenes observed by the two instruments. In particular, heating of the planetary boundary layer (PBL) and convection is likely to be more active during the daytime, which means that comparisons of the daytime measurements are more likely to show fine scale differences arising from natural variability.

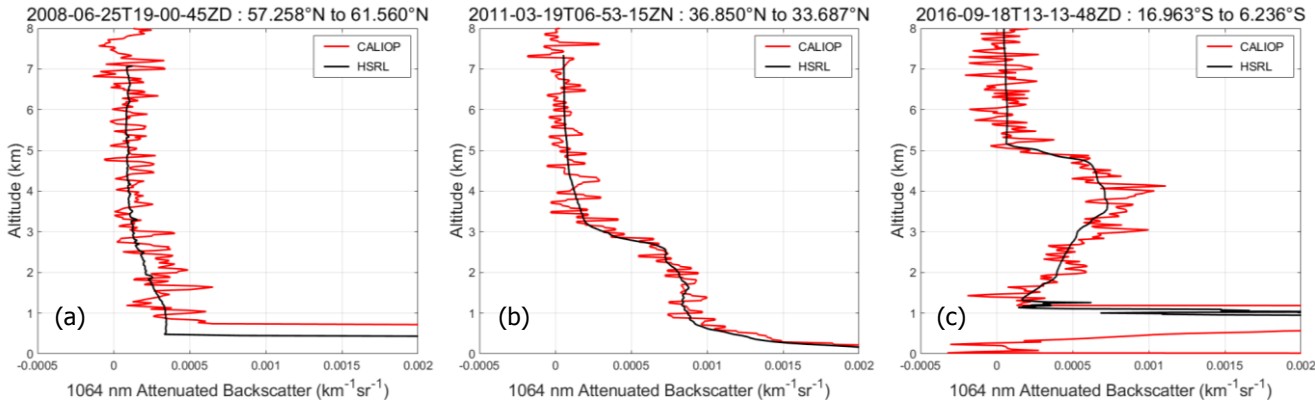

20 **Figure 28: comparisons of collocated HSRL and V4 CALIOP profiles of 1064 nm attenuated backscatter from (a) the ARCTAS2 field campaign in Fairbanks, Alaska during the summer of 2008; (b) the DISCOVER-AQ field campaign in the San Joaquin Valley of California during the spring of 2011; and (c) the ORACLES field campaign operated out of Walvis Bay, Namibia during the fall of 2016. CALIOP granule name identifiers are given in the plot titles.**

To derive quantitative comparisons of the CALIOP and HSRL 1064 nm measurements, we restricted the data from both

25 instruments to cloud-cleared profiles only, and these cloud-cleared profiles were averaged over the full extent of the coincident flight paths. The HSRL profiles are cloud-cleared using a hybrid approach that combines edge enhancement




using Haar wavelets with a more traditional thresholding technique (Burton et al., 2010). Cloud-clearing of the CALIOP L1 profiles is accomplished by inspecting the V4 L2 5 km merged layer products. These products report all cloud and aerosol layers detected at CALIOP's standard 5 km, 20 km and 80 km averaging resolutions, and also contain a complete record of all layers detected at single shot resolution (Vaughan et al., 2018). All profiles in which clouds or aerosols were detected

above the HSRL data acquisition altitude were excluded from the CALIOP averages. Doing this eliminates uncertainties that would otherwise be introduced by the attenuation corrections that would need to be applied to the CALIOP data acquired beneath the HSRL. Those CALIOP profiles in which clouds were detected between the HSRL flight altitude and the Earth's surface were also excluded from the data averaging process. Note that the aerosol-laden profiles shown in the right panel of Figure 28 were acquired while flying over an extended and continuous stratus cloud deck, and thus these data

were excluded from the comparisons.

The effects of this aggressive cloud-clearing strategy on the quality of the CALIOP averaged 1064 nm profiles are illustrated in Figure 29. Figure 29(a) shows the CALIOP 1064 nm attenuated backscatter measurements for a nighttime validation flight departing from Saint George's, Bermuda (32.36° N, 64.68° W) on 16 June 2014. Intermittent and occasionally opaque cirrus clouds are seen at and above the ~ 9 km HSRL flight altitude. A substantial cluster of midlevel clouds is also seen at ~

33° N in the CALIOP image. The presence of these clouds is reflected in the segments of completely missing data that appear centered near ~ 32° N in the HSRL image shown in Figure 29(c). Additional cloud-obscured regions of 'missing data' extend down to the surface from between ~ 3.5 km to ~ 2.5 km over most of the second half of the HSRL flight. Figure 29(b) shows the average of the 532 nm attenuated backscatter coefficient profiles taken over the full validation flight segment. Except for a small gap at ~ 6.5 km, this flight segment is, on average, cloudy from 12 km down to the surface.

Averages of the attenuated backscatter profiles that remained after application of the cloud clearing operations are shown in Figure 29(d) (CALIOP in red, HSRL in black). Of the 1500 laser pulses emitted by CALIOP during this HSRL underflight, only 386 were found to be entirely cloud-free. As a consequence, the SNR of the mean CALIOP profile is only ~ 1.3 times higher than would be typical for 80 km averaged profile acquired during nighttime operations. Furthermore, one of the potential pitfalls of different data acquisition times and data averaging volumes is clearly illustrated by the differences in

aerosol scattering magnitudes seen at ~ 0.4 km, where the CALIOP measurement is larger than HSRL by ~ 50 %.





**Figure 29:** nighttime CALIOP and HSRL measurements acquired 16 June 2014 over the Atlantic Ocean in the vicinity of Bermuda. The CALIOP profiles in panel (a) are averaged to a 5 km horizontal resolution (15 laser pulses acquired over ~ 0.75 s). The HSRL profiles shown in panel (c) are averaged to a temporal resolution of 10 seconds. The red lines at the bottom of panels a and c indicate coincident portions of the two data sets. Panel (b) shows the average of all CALIOP 532 nm attenuated backscatter coefficient profiles acquired during full validation flight segment while panel (d) shows the average of the CALIOP 1064 nm attenuated backscatter coefficient profiles that were classified as being cloud-free (386 of 1500).

To create comparable averages for both instruments over all flights, we corrected the CALIOP signal for the additional molecular attenuation incurred between CALIOP and the HSRL 1064 nm calibration altitude. No correction was applied for the attenuation due to undetected cloud or aerosol layers that may have been present between the two sensors. To date there have been 122 HSRL underflights of CALIPSO. After eliminating obvious instrument and data processing artifacts (e.g., those flights in which the CALIOP automated layer detection algorithm failed to identify either weak clouds or partially filled footprints, as evidenced by excessively large scattering ratios at 1064 nm, or for which the HSRL failed to acquire 1064 nm profile data), we obtained a data set consisting of 101 pairs of spatially collocated attenuated backscatter profiles. To compare these, we computed the ratios of the integrated attenuated backscatter coefficients (i.e., $\gamma'_{1064,CALIOP} / \gamma'_{1064,HSRL}$),





beginning at 6 km above mean sea level and extending downwards over successively larger altitude ranges of 2 km, 3 km, 4 km, and 5 km. Table 4 summarizes the results separately for nighttime and daytime measurements. The mean ratios over the higher altitude ranges – i.e., 6 km to 4 km and 6 km to 3 km – agree to within ±2 %, although, as should be expected, the standard deviations about the means are relatively high. As we proceed from these generally "clear air" regions deeper into

the PBL, the quality of the agreement deteriorates. From 6 km to 2 km, when the nighttime boundary layer is quiescent, the agreement between the two sets of measurements is still within ±2 %. However, during the daytime, when the PBL becomes more turbulent, the agreement falls off to within 5 %. Integrating from 6 km to 1 km produces the worst agreements: no closer than 11 % at night and 23 % during the day. We attribute these increasing disparities to the increasing non-uniformity of the PBL, both vertically and horizontally, as we extend our measurement range closer to the Earth's surface.

**Table 4: means (μ) and standard deviations (σ) of the ratios of integrated attenuated backscatters (CALIOP / HSRL) for 101 HSRL underflights of CALIPSO. Details about flight locations and dates can be found in Rogers et al., 2011 (day and night), Kar et al., 2018 (nighttime only), and Getzewich et al., 2018 (daytime only).**

| range | μ (night) | σ (night) | μ (day) | σ (day) |
|-------|-----------|-----------|---------|---------|
| 6 km to 4 km | 0.981 | 0.307 | 0.988 | 0.541 |
| 6 km to 3 km | 0.990 | 0.310 | 1.000 | 0.439 |
| 6 km to 2 km | 1.018 | 0.355 | 1.052 | 0.402 |
| 6 km to 1 km | 1.111 | 0.657 | 1.233 | 0.371 |

## 7    Discussion and concluding remarks

In this paper we describe the new techniques implemented in the CALIPSO version 4 (V4) data release to more accurately

calibrate the CALIOP 1064 nm measurements. There are two major differences between the version 3 (V3) and V4 calibration methods. First, the new cloud selection criteria implemented in V4 identify a much more homogeneous population of cirrus, while simultaneously weeding out the water clouds and polar stratospheric clouds that occasionally contaminated the V3 calibration procedure. Second, the data averaging scheme used to generate estimates of the calibration scale factors has been radically restructured. In previous versions of the 1064 nm calibration algorithm, all scale factors

accumulated over a full daytime or nighttime granule were averaged to create a single mean value that was applied everywhere within the granule. In contrast, the V4 algorithm accumulates scale factors in discrete increments of granule-elapsed time across multiple orbits. Doing this accommodates the substantial intra-orbit changes that are now known to characterize the scale factor time series. As a result of these changes, the calibration coefficients produced by the V4 algorithm can vary by ± 25 % relative to the V3 values and show distinctly different geospatial patterns that more accurately

reflect the continuously changing thermal environment onboard the satellite.





Despite the many significant differences, some critically important similarities remain between V3 and V4. As in previous versions of the CALIOP 1064 nm calibration algorithm, the fundamental assumption on which the V4 scheme relies is that the 1064 nm-to-532 nm ratio of particulate backscatter coefficients (i.e., the backscatter color ratio) for a specific subset of cirrus clouds is $1.01 \pm 0.25$. We assess the validity of this assumption and quantify the potential bias errors in the resulting

calibration coefficients by comparing the scale factors and calibration coefficients derived using cirrus clouds to those to those retrieved using ocean surfaces and water clouds as alternative calibration targets. These comparisons generally yield uniformly consistent results: the CALIOP calibration coefficients using the cirrus cloud technique typically lie within ~2 % or less of those derived using other calibration targets. The sole exception is our attempt to use opaque water clouds to calibrate the CALIOP nighttime measurements. The 1064 nm nighttime calibration coefficients derived using the water

cloud calibration method are, on average, larger than the cirrus-derived calibration coefficients by ~ 7 %. Furthermore, applying this same technique to nighttime 532 nm data produced 532 nm calibration coefficients that were ~ 13 % larger than the 532 nm calibration coefficients derived using the well-established molecular normalization method. Since the accuracy of 532 nm nighttime calibration coefficients derived using the molecular normalization technique has been firmly established using collocated high spectral resolution lidar (HSRL) measurements, we conclude that the non-ideal detector

response at 532 nm, when coupled with V4's overly-aggressive layer base detection algorithm, currently eliminates opaque water clouds as suitable targets for direct calibration of the CALIOP nighttime measurements at both 532 nm and 1064 nm.

The calibration comparison studies described in Sect. 6 demonstrate that the attenuated backscatter coefficients derived from the V4 CALIOP 1064 nm calibration procedures are both internally consistent and externally consistent. To be internally consistent, the attenuated backscatter coefficients measured within layers other than cirrus clouds should very closely match

theoretical expectations derived for other well-characterized atmospheric features. Whenever possible, these expectations should be based on stable, intrinsic properties that depend on layer type only, and not on highly variable extrinsic properties such as backscatter intensity. When the appropriate corrections for multiple scattering are made, we derive backscatter color ratios from calibrated measurements of opaque water clouds that very closely match the theoretically expected values (Sect. 6.4), and thus solidly establish the internal consistency of the cirrus cloud calibration technique. External consistency is best

demonstrated by traditional lidar validation studies, wherein coincident measurements are compared to those obtained using a previously authenticated instrument. We assess this by comparing the 1064 nm attenuated backscatter profiles reported in the V4 CALIOP level 1 data products to collocated, independently measured profiles of 1064 nm attenuated backscatter coefficients acquired by the NASA-LaRC HSRLs (Sect. 6.6), which currently provide the most reliable validation measurements available to space-based lidars. For measurements acquired in stable atmospheric conditions, the CALIOP

1064 nm attenuated backscatter coefficients lie typically within ~2% or less of those acquired by the LaRC HSRLs.

Based on a full year of data, median random uncertainties in the individual V4 1064 nm calibration coefficients are estimated at 1.63 % with a spread (median absolute distance) of $\pm$ 0.29 % at night, and 1.77 % $\pm$ 0.41 % during the day (Sect. 4.3). When considering both random and bias errors, the accuracy of the CALIOP 1064 nm calibration is expected to be within



3% both daytime and nighttime for a large majority of the measurements. This significant reduction in the CALIOP 1064 nm calibration coefficient uncertainty yields greatly improved reliability in several important downstream CALIOP data analyses such as cloud-aerosol discrimination and the retrieval of 1064 nm aerosol optical depths.

**Data Availability:** The CALIPSO lidar data products are available at the Atmospheric Science Data Center at NASA LaRC
(https://eosweb.larc.nasa.gov/project/calipso/calipso_table) and at the AERIS/ICARE Data and Services Center (http://www.icare.univ-lille1.fr). The CALTRACK data are available via AERIS/ICARE. HSRL data are available by request from the authors (John Hair at johnathan.w.hair@nasa.gov). MODIS atmospheric data (doi:10.5067/MODIS/MYD04_L2.061) are distributed via the MODIS web site at https://modis-atmos.gsfc.nasa.gov/MOD04_L2/index.html.

**Competing Interests:** The authors declare that they have no conflicts of interest. Coauthor Jacques Pelon is a one of the guest editors for the CALIPSO Version 4 Algorithms and Data Products special issue in Atmospheric Measurements Techniques but did not participate in any aspects of the editorial review of this manuscript.

**Acknowledgements:** This paper is dedicated to the memory of William H. (Bill) Hunt: lidar pioneer, instrument designer extraordinaire, gentleman, mentor, and friend. No man or woman on the planet contributed more to the success of the
CALIPSO lidar than Bill Hunt.

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
