# Peer review of "CALIPSO Lidar Calibration at 1064 nm: Version 4 Algorithm"

_Atmospheric Measurement Techniques, 2018_

## Referee Comment (RC1) · Anonymous Referee #1 · 11 Oct 2018

The authors present an excellent, comprehensive paper detailing a revised, new version, V4, data processing/assessment procedure for significantly improving the calibration of the 1064 nm channel of CALIOP. They present a detailed assessment of the deficiencies of the current version, V3, calibration procedure, followed by a detailed presentation of the new, V4, calibration procedure that discusses how the deficiencies of the V3 procedure are removed with the new V4 procedure. A key aspect of the V4 procedure is the selection of quality cirrus clouds for transferring the 532 nm calibration to 1064 nm. The authors present detailed assessments of data examples which support the new procedure for selecting quality cirrus clouds for the calibration transfer. The new procedure is validated by comparison with alternate calibration procedures (i.e., using dense water clouds, ocean surface lidar returns and attenuated

backscatter comparisons with nearly concurrent airborne High Spectral Resolution Lidar (HSRL) observations). They conclude that the cirrus cloud calibration approach as implemented with the V4 procedure is the best of these various calibration approaches in both providing the most coverage over orbits and estimated relative calibration accuracy (within $\sim 3\%$). With this level of accuracy, the 1064 nm attenuated backscatter data can now be used for a variety of quantitative applications of the CALIOP 1064 nm observations.

---

## Referee Comment (RC2) · Anonymous Referee #2 · 9 Nov 2018

The paper is very well written by a lidar expert team. An exhausting presentation and discussion about the calibration of CALIOP's 1064 nm channel is provided. I enjoyed reading very much and learned a lot. Even for ground-based lidar teams working with 532 and 1064 nm channels the paper is very valuable. Many new aspects are included in the V4 calibration scheme, compared to the V3 scheme. A very detailed description is given.

I was not able to find any point of criticism. And this very unusual for me, working as a reviewer since more than 25 years.

Publish as is!
* * *

---

## Author Comment (AC1) · 20 Nov 2018

Referee comments in black
Author responses in blue

We extend our deepest thanks to the referees, first for their careful reading of our (dauntingly long!) manuscript, and second for their very gracious remarks about the quality and future utility of our work. Developing the CALIOP 1064 nm calibration algorithms is very much a team effort, and it is extremely gratifying to have the expertise of our team members publicly recognized by the AMT referees.

**Interactive comments on "CALIPSO Lidar Calibration at 1064 nm: Version 4 Algorithm" by Mark Vaughan et al.**

Anonymous Referee #1

The authors present an excellent, comprehensive paper detailing a revised, new version, V4, data processing/assessment procedure for significantly improving the calibration of the 1064 nm channel of CALIOP. They present a detailed assessment of the deficiencies of the current version, V3, calibration procedure, followed by a detailed presentation of the new, V4, calibration procedure that discusses how the deficiencies of the V3 procedure are removed with the new V4 procedure. A key aspect of the V4 procedure is the selection of quality cirrus clouds for transferring the 532 nm calibration to 1064 nm. The authors present detailed assessments of data examples which support the new procedure for selecting quality cirrus clouds for the calibration transfer. The new procedure is validated by comparison with alternate calibration procedures (i.e., using dense water clouds, ocean surface lidar returns and attenuated backscatter comparisons with nearly concurrent airborne High Spectral Resolution Lidar (HSRL) observations). They conclude that the cirrus cloud calibration approach as implemented with the V4 procedure is the best of these various calibration approaches in both providing the most coverage over orbits and estimated relative calibration accuracy (within ~ 3%). With this level of accuracy, the 1064 nm attenuated backscatter data can now be used for a variety of quantitative applications of the CALIOP 1064 nm observations.

Anonymous Referee #2

The paper is very well written by a lidar expert team. An exhausting presentation and discussion about the calibration of CALIOP's 1064 nm channel is provided. I enjoyed reading very much and learned a lot. Even for ground-based lidar teams working with 532 and 1064 nm channels the paper is very valuable. Many new aspects are included in the V4 calibration scheme, compared to the V3 scheme. A very detailed description is given.

I was not able to find any point of criticism. And this very unusual for me, working as a reviewer since more than 25 years.

Publish as is!
* * *
Interactive comments on Atmos. Meas. Tech. Discuss., doi:10.5194/amt-2018-303, 2018.